# PTBP1 depletion in mature astrocytes reveals distinct splicing alterations without neuronal features

**Min Zhang[1,2†], Naoto Kubota[1,2†], David Nikom[1,2†], Ayden Arient[3], Sika Zheng[1,2]\***

[1]Division of Biomedical Sciences, University of California, Riverside, Riverside, United States; [2]Center for RNA Biology and Medicine, University of California, Riverside, Riverside, United States; [3]Department of Molecular, Cell and Systems Biology, University of California, Riverside, Riverside, United States

**\*For correspondence:**
sika.zheng@ucr.edu

[†]These authors contributed equally to this work

**Competing interest:** The authors declare that no competing interests exist.

## eLife Assessment

This study reports **important** negative results, showing that genetically removing the RNA-binding protein PTBP1 in astrocytes is insufficient to convert them into neurons, thereby challenging previous claims in the field. It also offers a **compelling** analysis of PTBP1's role in regulating astrocyte-specific splicing. The evidence is strong, as the experiments are technically sound, carefully controlled, and supported by both imaging and transcriptomic analyses.

**Abstract** Astrocyte-to-neuron reprogramming via depletion of PTBP1, a potent repressor of neuronal splicing, has been proposed as a therapeutic strategy, but its efficacy remains debated. While some reported successful conversion, others disputed this, citing a lack of neuronal gene expression as evidence of failed reprogramming. This interpretation was further challenged, attributed to incomplete PTBP1 inactivation, fueling ongoing controversy. Mechanistic understanding of the conversion, or the lack thereof, requires investigating, in conjunction with lineage tracing, the effect of *Ptbp1* loss of function in mature astrocytes on RNA splicing, which has not yet been examined. Here, we genetically ablated PTBP1 in adult Aldh1l1-Cre/ERT2 Ai14 mice to determine whether lineage-traced *Ptbp1* knockout astrocytes exhibited RNA splicing alterations congruent with neuronal differentiation. We found no widespread induction of neurons, despite a minuscule fraction of knockout cells showing neuron-like transcriptomic signatures. Importantly, PTBP1 loss in mature astrocytes induced splicing alterations unlike neuronal splicing patterns. These findings suggest that targeting PTBP1 alone is ineffective to drive neuronal reprogramming and highlight the need for combining splicing and lineage analyses. Loss of astrocytic PTBP1 is insufficient to induce neuronal splicing, contrasting with its well-known role in other non-neuronal cells, and instead affects a distinct astrocytic splicing program.

## Introduction

Neurons in the adult central nervous system (CNS) possess limited regenerative capacity, posing a great challenge in restoring brain function in neurodegenerative disease. The irreversible nature of CNS damage has spurred a wave of effort to convert non-neuronal cells into functional neurons to compensate for those lost in disease (*Bocchi et al., 2022*). The use of endogenous cells for direct neuronal conversion presents key advantages stemming from their proliferative capacity, maintenance of epigenetic and transcriptomic aging signatures lost upon reprogramming to stem cells, and bypassing a proliferative intermediate stage (*Mertens et al., 2015*; *Yang et al., 2015*). Diverse

neuronal conversion protocols targeting microglia (*Matsuda et al., 2019*), fibroblasts (*Torper et al., 2013*; *Yoo et al., 2011*), oligodendrocyte precursor cells (*Heinrich et al., 2014*), pericytes (*Karow et al., 2018*), and astrocytes *Mattugini et al., 2019* have shown feasibility of neural reprogramming. However, such approaches remain controversial for therapeutic application. *In vivo* neuronal reprogramming studies often achieve variable levels of success and provide inadequate evidence to explicitly show converted neurons originate from endogenous non-neuronal cells (*Liu et al., 2024*; *Qian et al., 2021*; *Rao et al., 2021*).

Recent reports have aimed to achieve neuronal reprogramming in the brain through depletion of polypyrimidine tract binding protein 1 (PTBP1) in glia (*Fu and Mobley, 2023*). PTBP1 is an RNA-binding protein that functions as a master repressor of neuronal splicing (*Keppetipola et al., 2012*; *Vuong et al., 2016b*; *Vuong et al., 2016a*; *Zheng et al., 2012*). To date, seven studies have reported the success of glial *Ptbp1* depletion to induce neuronal conversion *in vivo*. Initial observations by Weinberg et al. demonstrated an apparent generation of functional striatal neurons from oligodendrocytes targeted by an oligotropic adeno-associated viral vector (AAV) encoding *Ptbp1* siRNAs in wild-type rats (*Weinberg et al., 2017*). Qian et al. employed delivery of AAV vectors encoding *Ptbp1* shRNA as well as *Ptbp1* antisense oligonucleotides (ASO) into the substantia nigra of 6-hydroxydopamine-treated (6-OHDA) mice modeling Parkinson's disease (PD) and reported generation of new neurons and reversal of PD phenotypes (*Qian et al., 2020*). Zhou et al. delivered CRISPR-CasRx guide RNAs targeting *Ptbp1* mRNA to 6-OHDA mouse striatum and reported induction of dopaminergic neurons and rescue of motor deficits (*Zhou et al., 2020*). Maimon et al. used intracerebroventricular (ICV) delivery of anti-*Ptbp1* ASO to generate new neurons in aged wild-type mouse hippocampus and improve cognitive measures (*Maimon et al., 2021*). Yang et al. reported neuronal reprogramming of astrocytes in a mouse model of spinal cord injury by viral GFAP promoter-driven *Ptbp1* shRNA administration and accompanying recovery of motor function (*Yang et al., 2023a*). Yuan et al. used viral delivery of GFAP promoter-driven *Ptbp1* shRNA into the cortex of an ischemic stroke mouse model to induce astrocyte-to-neuron conversion and reported brain tissue repair of the infarction site (*Yuan et al., 2024*). Fukui et al. virally transduced astrocytes with *AAV-pGFAP-CasRx-SgRNA-Ptbp1* by tail vein injection into adult mice following ischemic stroke and reported generation of neurons in the dentate gyrus and rescue of memory deficits (*Fukui et al., 2024*). However, some studies have called this reprogramming approach into question. For example, Guo et al. failed to detect converted hippocampal, striatal, or substantia nigral neurons upon *AAV-GFAP-shPtbp1* injection in both wild-type and modeled Alzheimer's disease mice (*Guo et al., 2022*).

These studies exercised a variety of *Ptbp1* targeting approaches across a range of disease models but did not demonstrate that newly converted neurons truly arose from resident glial cells. Subsequent studies have conducted lineage tracing experiments to investigate the effects of *Ptbp1* loss on glia-to-neuron conversion. Wang et al. delivered AAVs encoding *Ptbp1* shRNA or *AAV-GFAP-CasRx-Ptbp1* and a fluorescent reporter to the striatum of an astrocyte lineage-tracing mouse model and found no co-labeling of lineage-traced astrocytes with *Ptbp1*-depleted neurons (*Wang et al., 2021*). Chen et al. employed Aldh1l1-Cre/ERT2 mice bred with hemagglutinin (HA) reporter mice to lineage trace reactive astrocytes in the substantia nigra or striatum of 6-OHDA mice treated with *AAV-shPtbp1* or *Ptbp1* ASO and observed no neuronal conversion in traced glia (*Chen et al., 2022*). Hoang et al. found similar results in the mouse retina upon genetic *Ptbp1* knockout (KO) by tamoxifen induction in GLAST-Cre/ERT2; Sun1-GFP$^{loxP/loxP}$; *Ptbp1*$^{loxP/loxP}$ mice in Müller glia, with only subtle changes in gene expression by single-cell RNA-sequencing (scRNA-seq) analysis of *Ptbp1*-depleted Müller glia (*Hoang et al., 2022*). A follow-up study using Aldh1l1-Cre/ERT2; Sun1-GFP$^{loxP/loxP}$; *Ptbp1*$^{loxP/loxP}$ mice provided identical results by 2, 4, and 8 weeks following Cre induction in mouse cortex, striatum, and substantia nigra (*Hoang et al., 2023*). Taken together, these independent studies have cast doubt on *Ptbp1* loss-of-function approaches for neuronal reprogramming *in vivo*.

Counterarguments against the lack of astrocyte-to-neuron conversion have emerged, concerning the degree of PTBP1 depletion and inconsistency between approaches (*Hao et al., 2023*). First, insufficient knockdown of *Ptbp1* using RNAi or CRISPR systems may result in ineffective loss of function. Second, durations following PTBP1 depletion do not align precisely between studies. Even *Hoang et al., 2023* investigation using a genetic *Ptbp1* knockout was challenged by *Hao et al., 2023* for inefficiencies, who attributed the lack of gene expression changes based on the scRNA-seq data obtained at 4 weeks after 4-OHT induction of knockout to ineffective PTBP1 depletion.

The controversy centers on the interpretation of the absence of gene expression changes: Hao et al. interpreted this as an insufficient loss of PTBP1 function, while Hoang et al. viewed it as a failure of astrocyte-to-neuron conversion. Indeed, no prior studies on astrocyte-to-neuron conversion, including those reporting positive conversion, have investigated PTBP1's well-established molecular function of regulating alternative splicing.

Thus, three key issues remain concerning the effectiveness of adult *Ptbp1* loss of function on neuronal reprogramming in the brain. First, no study has simultaneously examined the impact of *Ptbp1* loss on both alternative splicing and cellular changes, which is needed to firmly establish, or rule out, a direct relationship between PTBP1's molecular function and its influence on neuronal reprogramming. The downstream effect of *Ptbp1* loss on alternative splicing in astrocytes remains unknown. Second, additional *Ptbp1* genetic mutants are needed to rigorously evaluate *Ptbp1* loss of function *in vivo* as only one previous study has used genetic deletion. This is necessary because Hao et al. argued that complete deletion of PTBP1 induced cell death and there could be off-target effects of various knockdown approaches in prior publications. Third, a thorough investigation of transcriptomic changes upon astrocytic *Ptbp1* knockout for a longer period matching those in reports of positive conversion is needed for a fair comparison.

Here, we address both the RNA splicing and cell type changes following PTBP1 depletion in mature astrocytes using astrocyte-specific inducible *Ptbp1* conditional knockout (*Ptbp1* cKO) mice carrying a different *Ptbp1* loxP allele than the one in Hoang's study (*Hoang et al., 2023*). We performed genetic lineage tracing and enriched the control and *Ptbp1* cKO astrocytes in adult *Ptbp1*$^{loxP/loxP}$;tdT$^{+/-}$;Aldh1l1-Cre$^{+/-}$mice for scRNA-seq analysis as well as for bulk RNA-seq and splicing analysis following *Ptbp1* depletion. We observed efficient depletion of PTBP1 from astrocytes but no neuronal conversion in *Ptbp1* cKO mice at 4, 8, and 12 weeks after *Ptbp1* depletion. Our RNA-seq analysis of *Ptbp1*-depleted astrocytes showed minimal alterations in astrocytic gene expression but obvious changes in RNA splicing profiles, confirming loss of PTBP1 function. However, these splicing changes do not resemble the characteristics typically observed in neuronal splicing. ScRNA-seq analysis further revealed limited astrocyte-to-neuron conversion following genetic *Ptbp1* loss. We conclude that *Ptbp1* depletion in mature astrocytes effectively alters RNA splicing but is insufficient to induce widespread neuronal conversion. Intriguingly, our findings also indicate that PTBP1 in mature astrocytes, unlike in other non-neuronal cell types, is a dispensable repressor of neuronal splicing and controls a distinct astrocytic splicing program.

## Results
### Establishing an astrocyte-specific *Ptbp1* conditional KO (cKO) mouse model

To specifically knockout *Ptbp1* in mature astrocytes, we crossed *Ptbp1*$^{loxP/loxP}$ mice containing loxP sites flanking *Ptbp1* exon 2 (*Stork et al., 2019*; *Yeom et al., 2018*) with the astrocyte-specific tamoxifen-inducible mouse line Aldh1l1-Cre/ERT2 (*Srinivasan et al., 2016*), and with a Cre-dependent tdTomato reporter line Ai14 (LSL-tdTomato-WPRE) (*Figure 1A*). This allows labeling of *Ptbp1* cKO astrocytes (*Ptbp1*$^{loxP/loxP}$;tdT$^{+/-}$;Aldh1l1-Cre$^{+/-}$) by the tdTomato reporter for downstream analyses. This *Ptbp1* loxP allele is different from the previous astrocyte-to-neuron conversion study (*Hoang et al., 2023*), in which the promoter and first exon were removed. The Cre-dependent reporter is also different from Hoang et al.'s study, which used LSL-Sun1-sfGFP. Therefore, our experiments provide an independent assessment of astrocyte-specific *Ptbp1* cKO and monitoring of cKO cells.

Since astrocyte maturation persists in the first month following birth (*Felix et al., 2021*; *Laywell et al., 2000*; *Moroni et al., 2018*; *Qian et al., 2000*), we administered tamoxifen by intraperitoneal (IP) injection for five consecutive days from postnatal day 35–39 (P35–39). We then collected and analyzed mouse brains at 4 weeks, 8 weeks, and 12 weeks after tamoxifen induction, following the schedule in the Hao et al. study that reported positive conversion (*Figure 1A*). We first analyzed the consistency of the Cre reporter expression between control and *Ptbp1* cKO mice. We found that Cre reporter-positive tdTomato-expressing cells made up approximately 6–7% of the cortical cell population 4 weeks after tamoxifen induction and persisted for the duration of the 12-week analysis window (*Figure 1B–D*). The ratio of tdT$^+$ to DAPI$^+$ cells in the cortex remained around 6% at 8 weeks and 12 weeks following tamoxifen induction and did not change between control and *Ptbp1* cKO mice

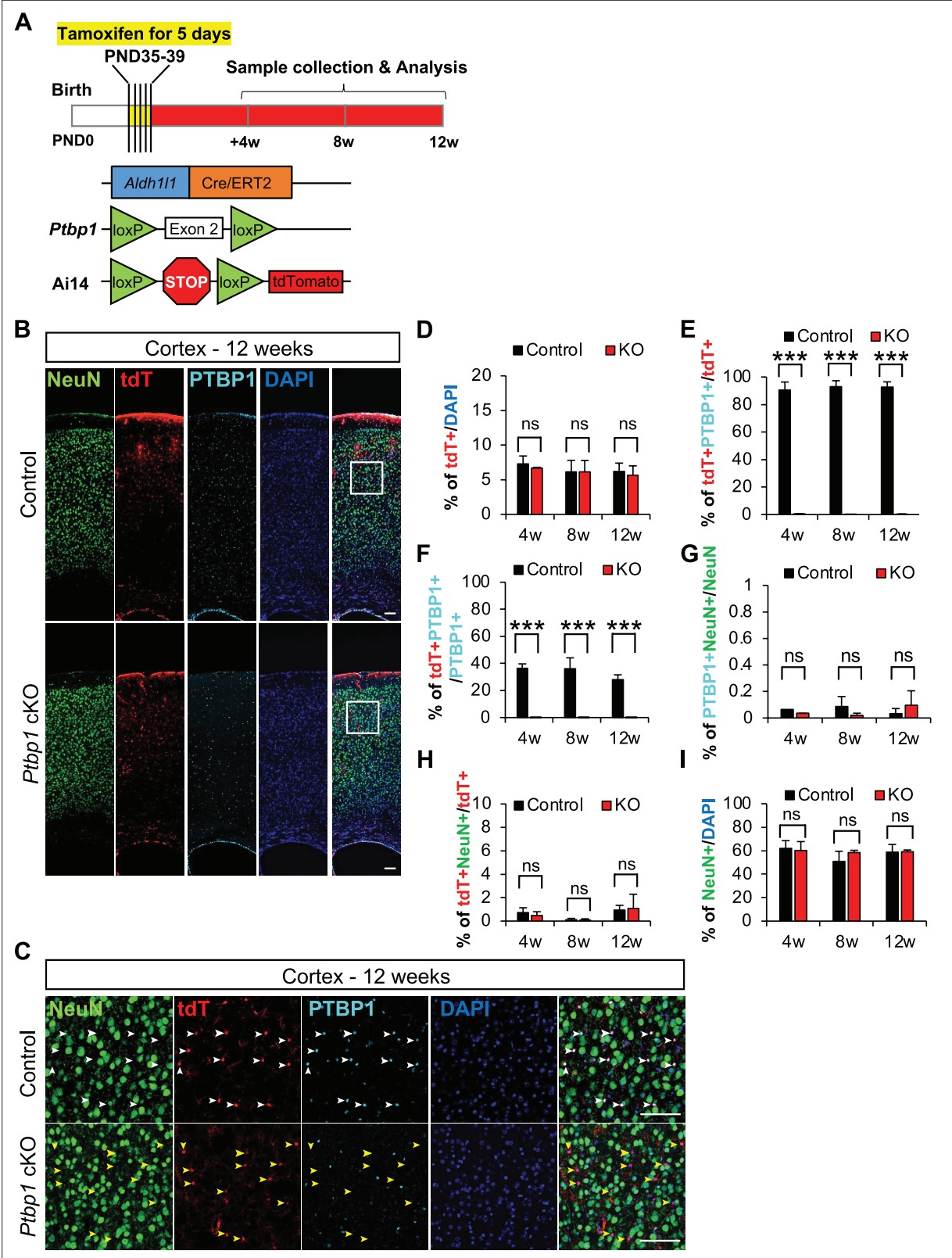

**Figure 1.** *Ptbp1* depletion does not effectively induce the astrocyte-to-neuron conversion in mouse cortex. (**A**) Schematic workflow to genetically generate the adult astrocyte specific *Ptbp1* conditional knockout (cKO) mouse model and timeline of tamoxifen administration and sample collection. (**B**) Representative images of mouse cortex collected 12 weeks after tamoxifen injection. White boxes indicate the location of images shown in (**C**) (scale bars are 100 µm). (**C**) Representative immunostaining images of mouse cortex collected 12 weeks after tamoxifen intraperitoneal (IP) injection. White

*Figure 1 continued on next page*

*Figure 1 continued*

arrowheads in the control panels indicate the expression of PTBP1 in tdTomato$^+$ (tdT$^+$) astrocytes. Yellow arrowheads indicate PTBP1 was successfully depleted in *Ptbp1* cKO astrocytes. The absence of NeuN and tdTomato double-positive (NeuN$^+$tdT$^+$) cells demonstrates no astrocyte-to-neuron conversion with *Ptbp1* depletion. Scale bars are 100 μm. (**D**) Quantification of tdT$^+$ astrocyte proportion at 4, 8, and 12 weeks following tamoxifen induction. (**E, F**) Quantification of *Ptbp1* knockout efficiency in control and *Ptbp1* cKO mouse cortex at 4, 8, and 12 weeks following tamoxifen induction. (**G**) Quantification of PTBP1$^+$NeuN$^+$ double-positive cells indicating PTBP1 is not expressed in neurons. (**H**) Quantification of NeuN$^+$tdT$^+$ cells indicating the absence of astrocyte-to-neuron conversion. (**I**) Quantification of NeuN$^+$ cells at 4, 8, and 12 weeks following tamoxifen induction showing no changes in the proportion of neurons in control or *Ptpb1* cKO cortex. Animal numbers are n=3 for both control and KO groups at all three time points. For quantification, the individual cortical images taken per brain are N=20–24 for 4 weeks, 7–10 for 8 weeks and 10–20 for 12 weeks. The quantification results represent the average and *stdev* of biological replicates (**n**). The significance test was carried out by *t*-test, *p<0.05, **p<0.01, ***p<0.001 and 'ns' means no difference with p>0.05.

The online version of this article includes the following source data and figure supplement(s) for figure 1:

**Source data 1.** Source data for *Figure 1D*.

**Source data 2.** Source data for *Figure 1E*.

**Source data 3.** Source data for *Figure 1F*.

**Source data 4.** Source data for *Figure 1G*.

**Source data 5.** Source data for *Figure 1H*.

**Source data 6.** Source data for *Figure 1I*.

**Figure supplement 1.** tdTomato expression in cortical and striatal astrocytes.

**Figure supplement 2.** PTBP1 is expressed in both astrocytes and microglia but not in neurons in adult mouse brain.

**Figure supplement 3.** PTBP1 depletion for 4 and 8 weeks does not induce astrocyte-to-neuron conversion in the cortex.

(*Figure 1D*). These results showed that the Cre activity, indicated by tdTomato expression, already reached a plateau by 4 weeks after tamoxifen injection and remained stable through 12 weeks. These findings demonstrate a reliable marker for tracing the targeted population and suggest negligible cell death in *Ptbp1* cKO cells. Almost all tdT$^+$ cells co-stained with astrocyte marker S100β, indicating the reliability of the tdTomato as an astrocyte reporter in this mouse model (*Figure 1—figure supplement 1*).

Immunostaining for PTBP1 showed 90–96% of cortical tdT$^+$ astrocytes are PTBP1$^+$ in *Ptbp1*$^{loxP/+}$;tdT$^{+/-}$;Aldh1l1-Cre$^{+/-}$ or *Ptbp1*$^{+/+}$;tdT$^{+/-}$;Aldh1l1-Cre$^{+/-}$ mice (*Figure 1E*). We found that less than 0.8% of tdT$^+$ astrocytes co-stained with PTBP1 in *Ptbp1* cKO mice, indicating efficient astrocyte-specific deletion of PTBP1 in the cortex (*Figure 1E*). The KO is specific to tdT$^+$ cells, as we observed no tdTomato expression in PTBP1$^+$ cells in the cKO animals (*Figure 1C*) and the percentage of tdT$^+$PTBP1$^+$ cells decreased from around 36% to 0.2% of the total PTBP1 cells (*Figure 1F*). These findings further confirm the mutually exclusive expression pattern of PTBP1 and tdTomato in *Ptbp1* cKO mice and high efficiency of *Ptbp1* cKO in tdT$^+$ cells of the cortex. Taken together, we find that *Ptbp1* cKO mice exhibit robust induction of astrocyte-specific reporter expression, efficient and specific PTBP1 depletion, making it a reliable model to study loss of PTBP1 function in mature astrocytes in the adult mouse brain.

### *Ptbp1* depletion does not induce the astrocyte-to-neuron conversion in the mouse cortex

A number of independent studies have claimed that knockdown of *Ptbp1* in mature astrocytes/Müller glia in mouse brain, retina, or spinal cord induces neuronal conversion accompanied by functional recovery as late as 12 weeks after PTBP1 knockdown (*Fukui et al., 2024*; *Maimon et al., 2021*; *Qian et al., 2020*; *Yang et al., 2023a*; *Yuan et al., 2024*; *Zhou et al., 2020*). Others have disputed these claims (*Chen et al., 2022*; *Guo et al., 2022*; *Hoang et al., 2022*; *Hoang et al., 2023*; *Leib et al., 2022*; *Wang et al., 2021*; *Xie et al., 2022*) arguing that observed conversion is biased by the use of less rigorous cell type-specific promoters (*Wang et al., 2021*). Only one group from the latter studies used a genetic *Ptbp1* KO mouse model to test astrocyte-to-neuron conversion upon *Ptbp1* loss and obtained negative results (*Hoang et al., 2022*; *Hoang et al., 2023*). The pro-conversion group challenged this study for its short observation window, that is, examining *Ptbp1* mutant mice only up to 8 weeks after 4-OHT induction rather than at 12 weeks.

Here, we employed an independent astrocyte-specific genetic *Ptbp1* cKO mouse model for this controversial topic. Among studies with positive reports, conversion was observed following AAV-mediated *Ptbp1* KD up to 12 weeks (*Qian et al., 2020*; *Xie et al., 2022*). For a fair comparison, we assessed the degree of neuronal conversion at 4, 8, and 12 weeks after tamoxifen injection to fully cover previously reported investigation windows. We confirmed the non-neuronal expression pattern of PTBP1 in the adult wild-type mouse brain, showing little PTBP1 co-staining with NeuN+ cells in the cortex, striatum, or hippocampus (*Figure 1—figure supplement 2*). We found similar results in control and *Ptbp1* cKO mice, with less than 0.1% of NeuN+ cells co-staining with PTBP1 in the cortex across the 4–12-week period (*Figure 1G*, *Figure 1—figure supplement 3*). In contrast, PTBP1 co-staining with S100β, GFAP, and Iba1 showed PTBP1 expression in both astrocytes and microglia (*Figure 1—figure supplements 2 and 3*). However, we observed very little astrocyte-to-neuron conversion (indicated by tdT+NeuN+ cells) upon *Ptbp1* cKO, even at 12 weeks after tamoxifen induction (*Figure 1H*). Consistent with the lack of noticeable astrocyte-to-neuron conversion, the proportion of cortical NeuN+ cells to total cell counts remained unchanged between control and *Ptbp1* cKO animals and throughout the 4–12-week period (*Figure 1I*).

### *Ptbp1* depletion does not induce the astrocyte-to-neuron conversion in the striatum

Reports of *Ptbp1* knockdown-induced glia-to-neuron conversion also included brain regions of the nigrostriatal pathway targeted for neuronal reprogramming in PD mouse models (*Qian et al., 2020*; *Zhou et al., 2020*). Therefore, we asked whether astrocyte-specific *Ptbp1* cKO can induce neuron conversion in the striatum at 4, 8, and 12 weeks following tamoxifen induction. We found persistent tdTomato reporter labeling of striatal astrocytes at 4, 8, and 12 weeks following tamoxifen induction (*Figure 2A and B*, *Figure 2—figure supplement 1*). The proportion of tdT+ cells in the striatum was slightly lower than in the cortex, comprising 4–5% of total cells at 4, 8, and 12 weeks in both control and *Ptbp1* cKO mice after tamoxifen administration (*Figure 2C*). We observed efficient PTBP1 depletion from striatal tdT+ cells in *Ptbp1* cKO mice (*Figure 2D*). Control mice showed PTBP1 expression in approximately 95% of tdT+ cells, whereas *Ptbp1* cKO animals showed minimal PTBP1 expression in tdT+ cells at any time point (*Figure 2D*, *Figure 2—figure supplement 1*). Approximately 25–30% of PTBP1+ cells expressed tdTomato in the control mouse striatum, which was reduced to nearly 0% in *Ptbp1* cKO mice (*Figure 2E*). Both control and *Ptbp1* cKO mice exhibited less than 0.1% of striatal NeuN+ cells expressing PTBP1 (*Figure 2F*). These findings in the striatum indicate that PTBP1 is efficiently depleted across multiple brain regions of *Ptbp1* cKO mice at 4, 8, and 12 weeks following tamoxifen induction.

We did not detect neuronal conversion of tdT+ cells in the striatum by 12 weeks (*Figure 2A and B*). Control mice exhibited less than 1% of tdT+NeuN+ cells across all time points, and *Ptbp1* cKO mice did not have a significant increase (*Figure 2G*). *Ptbp1* cKO mice showed a statistically significant decrease in tdT+NeuN+ cells at 4 weeks in the striatum, but the absolute changes remain small. We found a slight increase in the proportion of NeuN+ cells in *Ptbp1* cKO mouse striatum at 8 weeks, although this effect was not detectable at 4 or 12 weeks following tamoxifen induction (*Figure 2H*). Given the absence of reporter labeling in this altered neuronal population (*Figure 2G*), we do not attribute the increased ratio of NeuN+ to DAPI+ cells observed at 8 weeks in *Ptbp1* cKO mice to potential neuronal conversion of striatal astrocytes induced by *Ptbp1* depletion. We conclude that *Ptbp1* depletion for 4, 8, or 12 weeks is insufficient to induce efficient neuronal conversion of astrocytes in the striatum.

### *Ptbp1* depletion does not induce the astrocyte-to-neuron conversion in the substantia nigra

The substantia nigra is another region of the nigrostriatal pathway examined for the conversion of astrocytes to neurons after knockdown of *Ptbp1* (*Qian et al., 2020*). Loss of dopaminergic neurons in the substantia nigra is a pathological hallmark of PD that causes striatal dopamine deficiency (*Poewe et al., 2017*). Qian et al. found that following RNAi-mediated *Ptbp1* knockdown, 20% of reporter-labeled cells expressed NeuN at 3 weeks, 60% at 5 weeks, and approximately 80% at 10 weeks (*Qian et al., 2020*). By 12 weeks, around 35% of reporter-positive cells exhibited markers for dopaminergic neurons, including tyrosine hydroxylase (TH) (*Qian et al., 2020*).

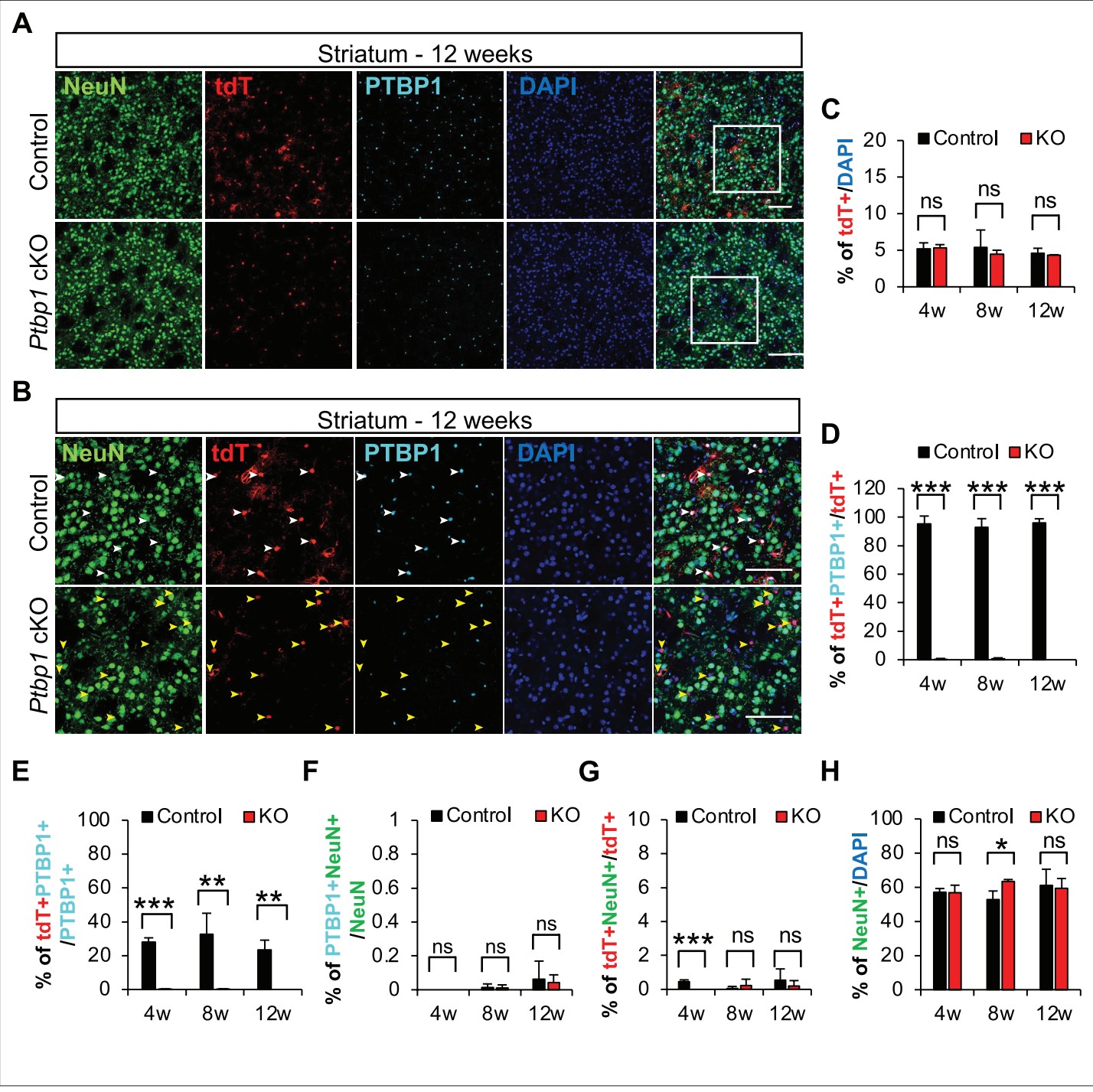

**Figure 2.** *Ptbp1* depletion does not induce the astrocyte-to-neuron transition in striatum. (**A**) Representative images of control and *Ptbp1* cKO mouse striatum collected 12 weeks after tamoxifen injection. White boxes indicate the location of images shown in (**B**). Scale bars are 100 μm. (**B**) Representative immunostaining images of mouse striatum collected 12 weeks after tamoxifen induction. White arrowheads in the control panels indicate the expression of PTBP1 in astrocytes (tdT+ cells). Yellow arrowheads indicate the efficient PTBP1 depletion in *Ptbp1* cKO astrocytes. The absence of NeuN+tdT+ cells in the striatum demonstrates no astrocyte-to-neuron conversion with *Ptbp1* depletion. Scale bars are 100 μm. (**C**) Quantification of striatal tdT+ astrocyte proportion at 4, 8, and 12 weeks following tamoxifen induction. (**D, E**) Quantification of *Ptbp1* knockout efficiency in control and *Ptbp1* cKO mouse striatum at 4, 8, and 12 weeks following tamoxifen induction. (**F**) Quantification of PTBP1+NeuN+ double-positive cells indicating PTBP1 is not expressed in striatal neurons. (**G**) Quantification of NeuN+tdT+ cells indicating absence of astrocyte-to-neuron conversion in the striatum. (**H**) Quantification of NeuN+ cells at 4, 8, and 12 weeks following tamoxifen induction showing minimal changes in the proportion of neurons in control or *Ptbp1* cKO striatum. Animal numbers are n=3 for both control and KO groups at all three time points. For

*Figure 2 continued on next page*

*Figure 2 continued*

quantification, the individual cortical images taken per brain are N=4–6 for 4 weeks, 3–6 for 8 weeks, and 4–6 for 12 weeks. The quantification results represent the average and *stdev* of biological replicates (**n**). The significance test was carried out by *t*-test, *p<0.05, **p<0.01, ***p<0.001 and 'ns' means no difference with p>0.05.

The online version of this article includes the following source data and figure supplement(s) for figure 2:

**Source data 1.** Source data for *Figure 2C*.

**Source data 2.** Source data for *Figure 2D*.

**Source data 3.** Source data for *Figure 2E*.

**Source data 4.** Source data for *Figure 2F*.

**Source data 5.** Source data for *Figure 2G*.

**Source data 6.** Source data for *Figure 2H*.

**Figure supplement 1.** 4 and 8 weeks of PTBP1 depletion do not induce astrocyte-to-neuron conversion in the striatum.

To determine if *Ptbp1* cKO mice show astrocyte-to-neuron conversion in the substantia nigra, we performed immunostaining on control and *Ptbp1* cKO samples for NeuN and TH. Both genotypes demonstrated a strong induction of tdTomato expression, which did not co-localize with NeuN or TH markers at 4, 8, or 12 weeks following tamoxifen treatment (*Figure 3A–C*). We confirmed PTBP1 depletion by immunostaining control and *Ptbp1* cKO samples for PTBP1, showing a similar pattern of PTBP1 loss in tdT+ cells as in the cortex and striatum (*Figure 3—figure supplement 1*). The lack of tdT$^+$NeuN$^+$ or tdT$^+$TH$^+$ cells in substantia nigra demonstrates that the depletion of *Ptbp1* does not facilitate the conversion of astrocytes to neurons, even after 12 weeks of *Ptbp1* deletion.

## Widespread splicing changes in *Ptbp1* cKO astrocytes

No studies have investigated the splicing landscape or confirmed the loss of PTBP1's splicing regulatory function in astrocytes depleted of PTBP1. To investigate gene expression and splicing changes resulting from genetic *Ptbp1* loss in astrocytes, we performed bulk RNA-seq on tdT$^+$ cells isolated from control and *Ptbp1* cKO animals. Cortices were isolated at postnatal week 9 from animals injected with tamoxifen at postnatal week 5, and tdT$^+$ cells were sorted by fluorescence-activated cell sorting (FACS) followed by bulk RNA-seq to obtain >100 million reads per sample (*Figure 4A*, *Figure 4—figure supplement 1*). RNA splicing analysis was conducted using the Shiba pipeline to explore splicing changes upon *Ptbp1* loss in astrocytes (*Kubota et al., 2025*).

Our analysis revealed a wide range of splicing changes, identifying 581 differentially spliced events (DSEs) in 467 genes across eight types of alternative RNA splicing patterns (*Figure 4B and C*). We confirmed that *Ptbp1* exon 2 was not expressed in the *Ptbp1* cKO samples (percent spliced in [PSI]=14.9), while it remained fully included in the control samples (PSI = 100), demonstrating the high KO efficiency of *Ptbp1* in astrocytes and the purity of the isolated tdT$^+$ cells (*Figure 4D*). Motif enrichment analysis revealed an overrepresentation of CU-rich sequences—a characteristic motif of PTBP binding sites (*Keppetipola et al., 2012*; *Vuong et al., 2016a*)—near the 3′ splice sites of differentially spliced skipped exons (*Figure 4E*), further validating the loss of PTBP1 function.

We then tested whether PTBP1 exhibits position-dependent regulation in adult astrocytes. We performed a stratified motif enrichment analysis based on the direction of regulation (activation or repression after PTBP1 loss) and motif position (upstream or downstream) (*Figure 4—figure supplement 2A*). CU-rich motifs were significantly enriched in the upstream introns of both activated and repressed exons upon PTBP1 loss, with higher enrichment observed in repressed exons (enrichment ratio = 2.14, $q=9.00 \times 10^{-5}$) compared to activated exons (enrichment ratio = 1.72, $q=7.75 \times 10^{-5}$) (*Figure 4—figure supplement 2B and C*). In contrast, no CU-rich motifs were found downstream of activated exons (*Figure 4—figure supplement 2D*), while a weak, non-significant enrichment was observed downstream of repressed exons (enrichment ratio = 1.21, $q=0.225$; *Figure 4—figure supplement 2E*). These results do not necessarily fully fit with a couple of earlier PTBP1 CLIP studies showing differential PTBP1 binding for repressed vs. activated exons but align with the Black Lab study that PTBP1 binds upstream introns of both repressed and activated exons. In either case, PTBP1 affects a diverse set of alternative exons and likely involves diverse context-dependent binding patterns.

Gene enrichment analysis for differentially spliced genes (DSGs) revealed enrichment in pathways related to generic transcription, chromatin organization, DNA damage checkpoint signaling, cell

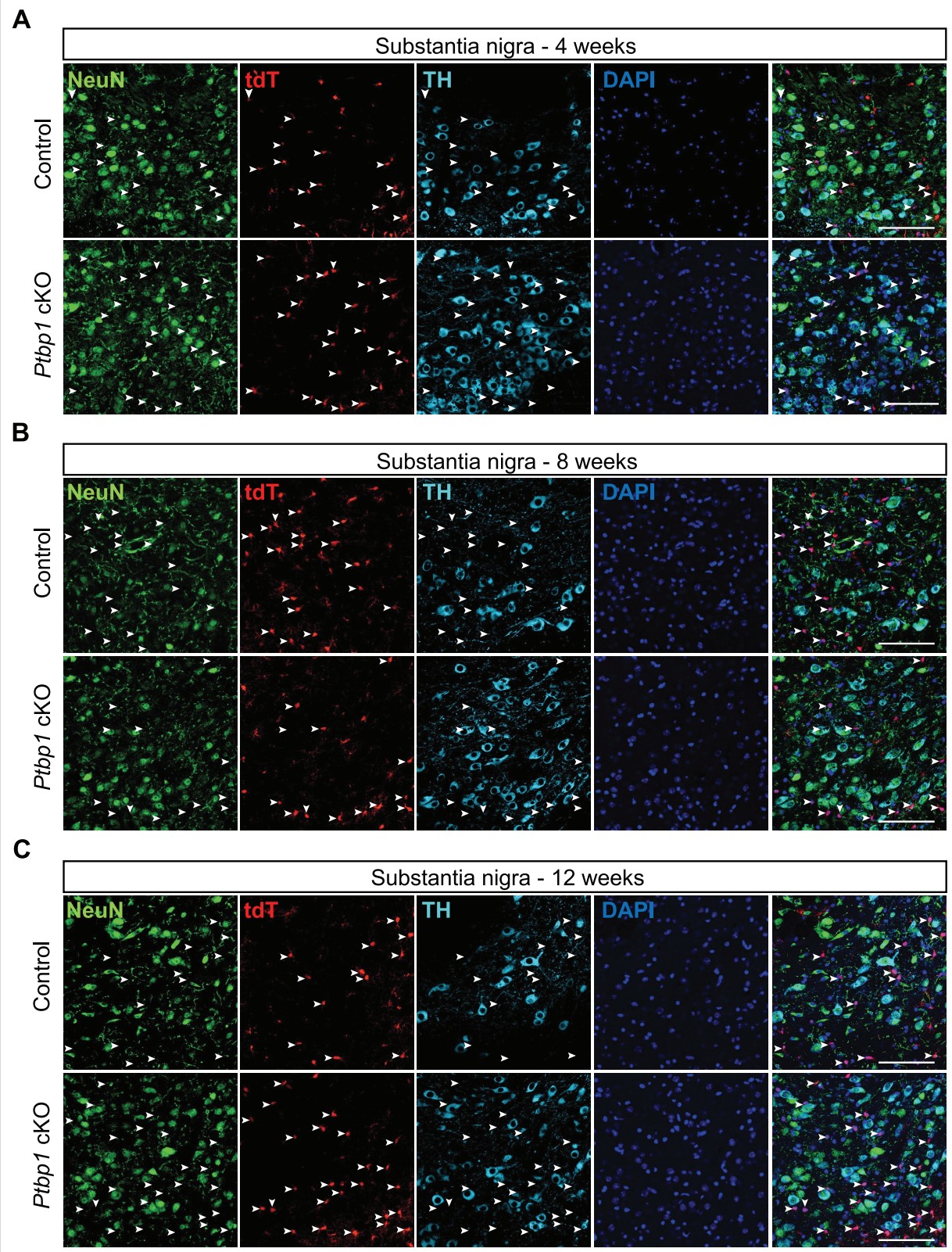

**Figure 3.** *Ptbp1* depletion does not induce the astrocyte-to-neuron transition in substantia nigra. (**A–C**) Representative images of the immunostaining results (substantia nigra) of the mouse brains collected at 4 weeks (**A**), 8 weeks (**B**), and 12 weeks (**C**) after tamoxifen induction. White arrowheads indicate the locations of astrocytes (tdT+ cells). However, none of the tdT+ cells express either NeuN or TH. The absence of NeuN or TH and tdTomato

*Figure 3 continued on next page*

*Figure 3 continued*

double-positive cells reveals no astrocyte-to-neuron conversion in *Ptbp1* cKO. Scale bars are 100 um. Animal numbers are n=3 for both control and KO groups at all the three time points except for the control group at 8 weeks (n=2).

The online version of this article includes the following figure supplement(s) for figure 3:

**Figure supplement 1.** PTBP1 depletion in substantia nigra 4 weeks after tamoxifen induction.

junction organization, and epigenetic regulation of gene expression (*Figure 4F*). Notably, no ontologies associated with neurogenesis or neuronal differentiation were significantly enriched, suggesting that *Ptbp1* loss in astrocytes does not drive a shift toward neuronal fate. This finding suggests that the splicing alterations are specific to astrocyte physiology and function rather than a change in cell fate, reinforcing the absence of astrocyte-to-neuron conversion upon genetic loss of *Ptbp1*.

## Splicing changes in *Ptbp1* cKO astrocytes do not resemble those associated with the acquisition of neuronal fates

To further investigate the relationship between splicing changes in *Ptbp1* cKO astrocytes and the splicing programs associated with the neuronal fate, we conducted principal component analysis (PCA) of the splicing profiles from the control and *Ptbp1* cKO astrocytes as well as different stages of *in vitro* neuronal differentiation. The results showed close clustering of the control and cKO astrocyte samples, which were positioned separately from *in vitro* differentiated neurons (*Figure 5A*). Interestingly, on PC1, the control and cKO astrocytes align with day *in vitro* (DIV) −4 and DIV0 cells, which are considered radial glia cells. PCA of the control and *Ptbp1* cKO astrocytes and developmental cortical tissue also showed the cKO samples do not deviate from the control astrocytes in a direction of neurogenesis (from E10 to P0, *Figure 5—figure supplement 1*).

We conducted pairwise comparisons of the splicing profiles between control and *Ptbp1* cKO astrocytes as well as those observed in DIV0 (radial glia) and DIV28 (maturing neurons). Correlation analysis demonstrated that *Ptbp1* cKO astrocytes retained a splicing profile very similar to the control astrocytes ( $\rho$ =0.92), and to a lesser degree DIV0 radial glia ( $\rho$ =0.80), but distinct from that of DIV28 neurons ( $\rho$ =0.24). Control astrocytes also appeared more similar to DIV0 radial glia than DIV 28 neurons ( $\rho$ =0.79 and 0.24 for DIV0 and DIV28, respectively) (*Figure 5B*). These results show that mature astrocytes exhibit splicing profiles different from radial glia or neurons but substantially closer to those of radial glia than neurons. More importantly, despite inducing significant splicing changes, PTBP1 loss has not forced astrocytes to adopt a neuronal splicing pattern. Pairwise comparisons using developmental cortical tissue show similar correlation results that the control and cKO astrocytes are more similar to E10 cortices than P0 cortices (*Figure 5—figure supplement 1*).

Next, we compared the splicing changes induced by PTBP1 loss and those by differentiation. The scatter plot of delta PSI (dPSI) in *Ptbp1* cKO astrocytes (cKO vs. Control) against the developmental splicing changes in *in vitro* neuronal differentiation (DIV28 vs. DIV0) showed that differentiation caused many more splicing alterations, most of which were not affected by PTBP1 depletion in astrocytes (F2 and F6 in *Figure 5C and D*). Events with positive correlation (F1 and F5) are not significantly more frequent than those with negative correlation (F3 and F7). F1 and F5 represent only 14.7% of total differential splicing events associated with neuronal differentiation, vs. 11.8% for F3 and F7, and the magnitudes of splicing changes induced by PTBP1 loss are substantially smaller than those occurring in differentiation. Indeed, only 17% of events in F1 and F5 are significantly different between *Ptbp1* cKO and control (|dPSI|>10, adjusted Fisher's exact test p<0.05, and Welch's *t*-test p<0.05). Therefore, PTBP1 depletion in mature astrocytes has caused an insignificant and incoherent effect of gaining characteristics of neuronal splicing. Similar results were observed when analyzing splicing changes during cortical development (*Figure 5—figure supplement 1*). In summary, splicing alterations upon *Ptbp1* deletion are primarily astrocyte-specific and do not align with the typical neuron-related splicing patterns observed during neuronal development.

To further validate that *Ptbp1* depletion does not induce a neuronal-like splicing program, we extended our comparison using an independent dataset of astrocyte-derived neurons generated through overexpression of proneural transcription factors (Ngn2 or mutant PmutNgn2) (*Pereira et al., 2024*). PCA using this dataset and ours showed that *Ptbp1* cKO astrocytes cluster tightly with control *in vivo* astrocytes across all components (*Figure 5—figure supplement 2A*). The *in vivo* splicing

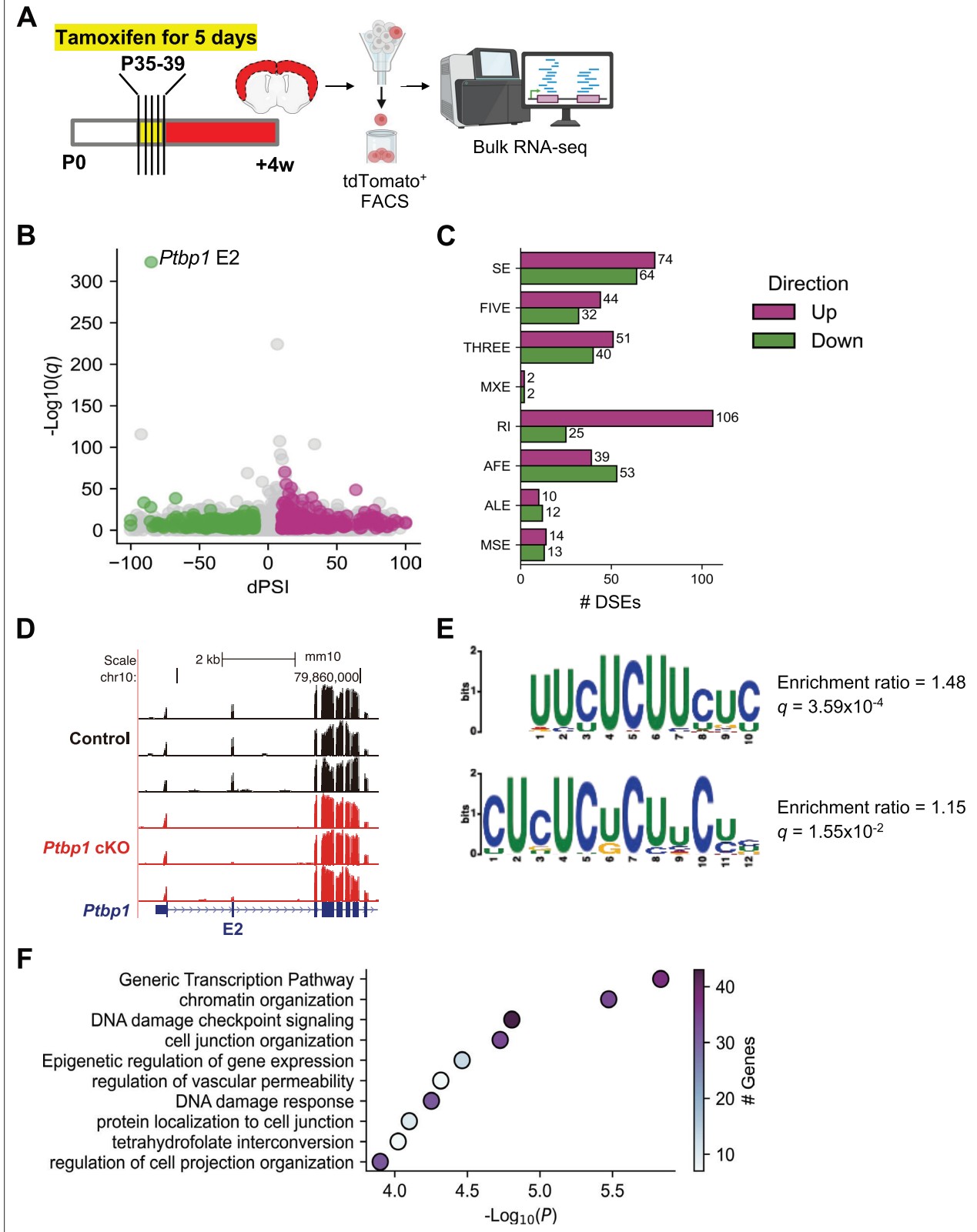

**Figure 4.** Widespread splicing changes in *Ptbp1* conditional knockout (cKO) astrocytes. (**A**) Schematic of the experimental design of bulk RNA-seq. (**B**) Volcano plot of differential splicing analysis, highlighting the significant exon 2 skipping in *Ptbp1* in the cKO samples. Upregulated events are highlighted in pink and downregulated events in green. (**C**) Bar plot showing the number of differentially spliced events (DSEs) identified in eight types of alternative splicing events. SE: skipped exon; FIVE: alternative 5′ prime splice site; THREE: alternative 3′ prime splice site; MXE: mutually exclusive

*Figure 4 continued on next page*

*Figure 4 continued*

exons; RI: retained intron; AFE: alternative first exon; ALE: alternative last exon; MSE: multiple skipped exons. (**D**) Genome browser track of the *Ptbp1* gene locus and its exon 2 (E2) with RNA-seq signals of control and *Ptbp1* cKO samples. (**E**) Enriched motifs in 3′ spliced sites of differentially spliced skipped exons by *Ptbp1* cKO. (**F**) Gene ontology enrichment analysis of differentially spliced genes in *Ptbp1* cKO astrocytes. The coronal section drawing in (**A**) was created using BioRender.com.

The online version of this article includes the following figure supplement(s) for figure 4:

**Figure supplement 1.** Fluorescence-activated cell sorting of tdTomato⁺ cells.

**Figure supplement 2.** Motif enrichment analysis for PTBP1-regulated exons.

profiles from this study and the *in vitro* splicing profiles from Pereira et al. are well separated on PC1 and PC2. In contrast, PC3 and PC4 revealed biologically meaningful separation between *in vitro* astrocytes and astrocyte-derived neurons. While Ngn2/PmutNgn2-induced neurons and GFP controls were distributed across a broader range along these axes, *Ptbp1* cKO samples remained tightly grouped with control astrocytes and showed no directional shift toward the neuronal cluster (*Figure 5—figure supplement 2B*). These findings further support the conclusion that PTBP1 depletion in mature astrocytes does not induce a neuronal-like splicing program, even when compared against neurons derived from the astrocyte lineage.

We next performed pairwise correlation analysis of PSI between *Ptbp1* cKO, control astrocytes, and PmutNgn2-induced neurons. These comparisons again confirmed that *Ptbp1* cKO astrocytes are highly similar to control astrocytes ($\rho$ =0.81) and clearly distinct from PmutNgn2-induced neurons ($\rho$ =0.62) (*Figure 5—figure supplement 2C*), reinforcing the notion that PTBP1 loss alone is insufficient to drive a neuronal-like splicing transition.

We plotted the splicing changes induced by PTBP1 depletion in astrocytes against those observed in PmutNgn2-induced neurons and classified events into eight categories based on the direction and significance of splicing changes (*Figure 5—figure supplement 2D*). The majority of events (F4 and F8) were significantly altered only in *Ptbp1* cKO astrocytes but not in PmutNgn2-induced neurons, while only a small fraction of events (F1 and F5) showed concordant changes in both conditions. The overlap of events with similar regulation (F1 and F5) was limited, comprising only 14.8% of total differentially spliced events (35/236) compared to 49.2% (116/236) in the *Ptbp1* cKO astrocytes-specific category (F4 and F8) (*Figure 5—figure supplement 2E*).

Together, these findings further support the conclusion that PTBP1 depletion in mature astrocytes does not recapitulate the splicing patterns associated with neuronal differentiation in the context of either developmental differentiation or *in vitro* astrocyte-to-neuron reprogramming. PTBP1-regulated splicing events in astrocytes reflect an astrocyte-intrinsic program, distinct from the broader remodeling observed during neuronal lineage acquisition.

## Thorough examination of gene expression changes in *Ptbp1* cKO astrocytes

The bulk RNA-seq data from purified tdT⁺ cells (astrocytes) provided the sequencing depth to thoroughly examine the gene expression changes after *Ptbp1* cKO. We found that only 11 genes were identified as differentially expressed genes (DEGs) between control and *Ptbp1* cKO samples (adjusted p<0.05 and |log2 fold change|>1) (*Figure 6A*). Among these, seven genes were upregulated: *Cfap54*, *Gm10722*, *Tdg-ps2*, *Zim1*, *Tmem72*, *Cpz*, and *Ntn4*. The four downregulated genes were *Tdg*, *1190007I07Rik/Brawnin*, *Cep83os*, and *Mt2*. Notably, none of the identified DEGs were markers of neurogenesis or any specific cell lineages, suggesting that *Ptbp1* loss does not induce an astrocyte-to-neuron conversion or trigger the expression of genes associated with neuronal lineages. Despite the deletion of exon 2 (*Figure 4D*), which introduces a frameshift predicted to disrupt the open reading frame and trigger nonsense-mediated mRNA decay (NMD), gene-level expression of *Ptbp1* remained largely unchanged between control and cKO samples (adjusted *P*=0.81, log2 fold change = –0.18) (*Figure 6A*). We suspect that the process of brain tissue dissociation and FACS sorting for bulk or single-cell RNA-seq may enrich for nucleic material and thus dilute the NMD signal, which occurs in the cytoplasm. Alternatively, the transcripts (like other genes) may escape NMD for unknown mechanisms. Although a frameshift is a strong indicator for triggering NMD, it does not guarantee NMD will occur in every case.

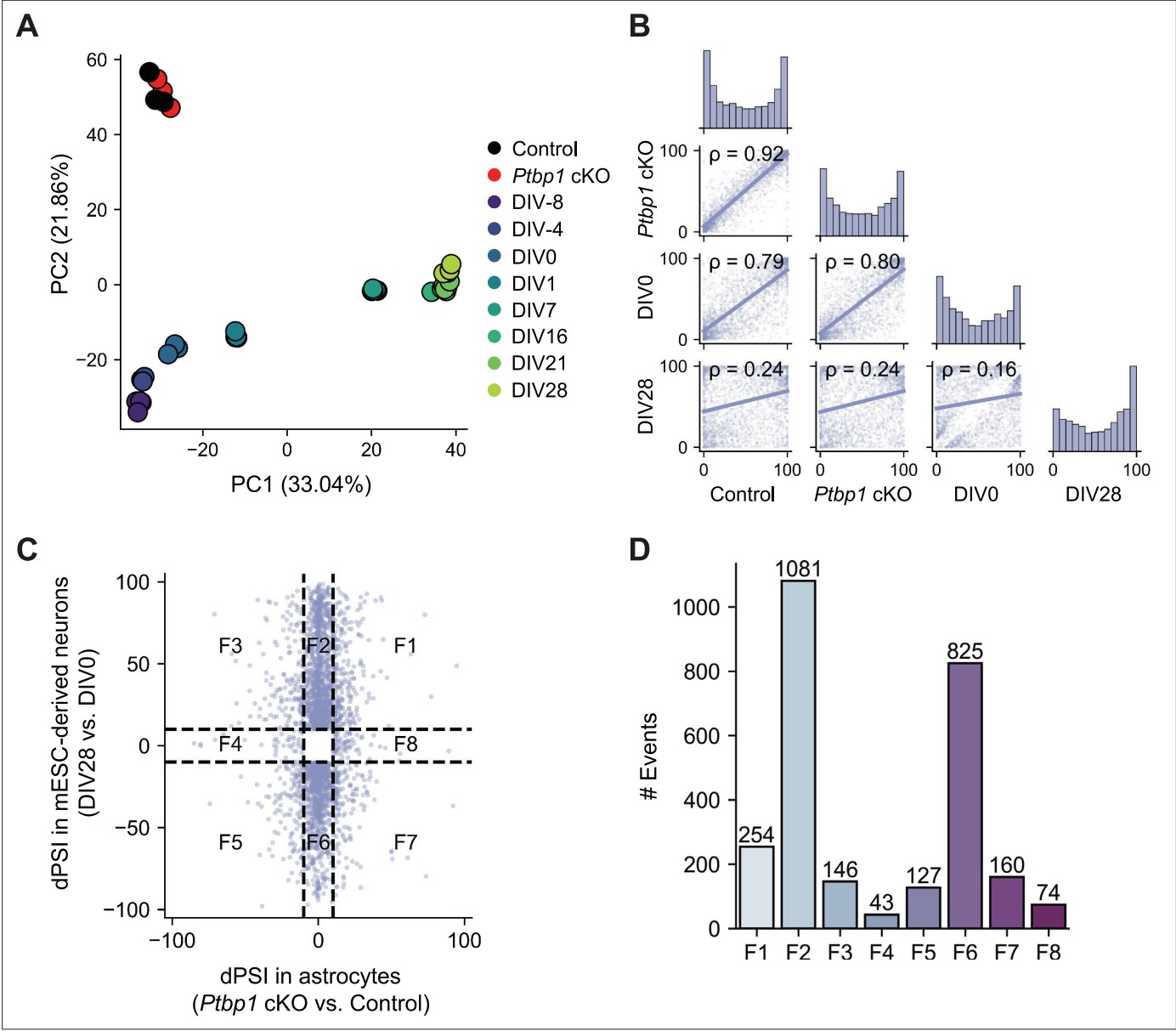

**Figure 5.** Impact of *Ptbp1* loss on astrocyte splicing profiles. (**A**) Principal component analysis (PCA) of percent spliced in (PSI) values across control, *Ptbp1* conditional knockout (cKO) astrocytes, and *in vitro* differentiated neuron samples at various differentiation stages (day *in vitro* (DIV) 8, DIV4, DIV0, DIV1, DIV7, DIV16, DIV21, and DIV28). (**B**) Spearman's correlation analysis of PSI values between control, *Ptbp1* cKO astrocytes, and *in vitro* differentiated neuron samples (DIV0 and DIV28). (**C**) Scatter plot of delta PSI (dPSI) in *Ptbp1* cKO astrocytes (cKO vs. Control) against splicing changes in *in vitro* differentiated neurons (DIV28 vs. DIV0). Splicing events are categorized into eight functional groups (F1–F8). (**D**) Bar plot showing the number of alternative splicing events in each functional category (F1–F8).

The online version of this article includes the following figure supplement(s) for figure 5:

**Figure supplement 1.** Comparison of splicing profiles between *Ptbp1* conditional knockout (cKO) astrocytes and developmental cortical tissue.

**Figure supplement 2.** Comparison of splicing profiles between *Ptbp1* conditional knockout (cKO) astrocytes and astrocyte-derived neurons.

The PCA of gene expression profiles, combined with RNA-seq data from the *in vitro* differentiated neurons, showed distinct clustering of control and cKO astrocytes from differentiating neurons (*Figure 6B*). The two astrocyte samples were closely grouped together and resembled DIV 0–1 cells, or radial glia, on PC1, but were positioned separately from all neuronal samples on PC2 (*Figure 6B*). Correlation analysis demonstrated almost identical gene expression profiles between control and *Ptbp1* cKO astrocytes ($\rho$ =0.99) (*Figure 6C*). Control astrocytes appeared more similar to DIV0 radial

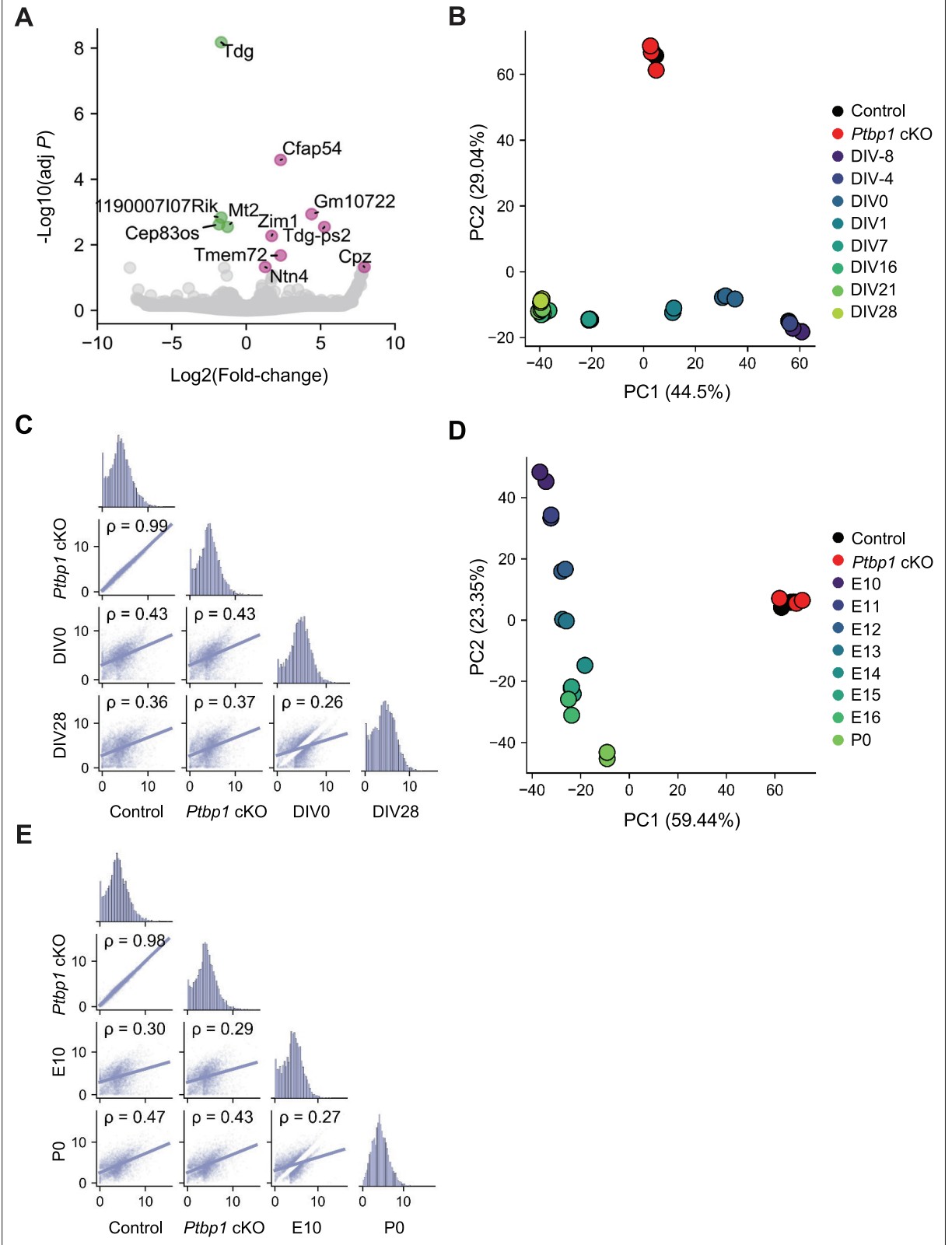

**Figure 6.** Minimal gene expression changes in *Ptbp1* conditional knockout (cKO) astrocytes. (**A**) Volcano plot showing differentially expressed genes (DEGs) between control and *Ptbp1* cKO astrocyte samples. Upregulated genes are highlighted in pink and downregulated genes in green. (**B**) Principal component analysis (PCA) of transcript per million (TPM) values across control, *Ptbp1* cKO astrocytes, and *in vitro* differentiated neuron samples at various differentiation stages (day *in vitro* (DIV) 8, DIV4, DIV0, DIV1, DIV7, DIV16, DIV21, and DIV28). (**C**) Spearman's correlation analysis comparing TPM

*Figure 6 continued on next page*

*Figure 6 continued*

values between control, *Ptbp1* cKO astrocytes, and *in vitro* differentiated neuron samples (DIV0 and DIV28). (**D**) PCA of TPM values across control, *Ptbp1* cKO astrocytes, and cortical tissue samples at various developmental stages (E10, E11, E12, E13, E14, E15, E16, and P0). (**E**) Spearman's correlation analysis comparing TPM values between control, *Ptbp1* cKO astrocytes, and cortical tissue samples (E10 and P0).

The online version of this article includes the following figure supplement(s) for figure 6:

**Figure supplement 1.** Comparison of gene expression profiles between *Ptbp1* conditional knockout (cKO) astrocytes and astrocyte-derived neurons.

glia than to DIV28 neurons ($\rho$ =0.43 and 0.36 for DIV0 and DIV28, respectively). This result highlights that mature astrocytes have more similarity to the expression profile of radial glia than neurons, as well as splicing profiles, but are still quite distinct. These patterns are not affected by PTBP1 depletion. Similar results were observed in samples of developmental cortical tissue, where PCA demonstrated clear separation between astrocytes and neuronal samples (***Figure 6D***), with no increase in correlation by PTBP1 loss (***Figure 6E***).

Consistent with this, PCA using an independent dataset of astrocyte-derived neurons (***Pereira et al., 2024***) showed that control and *Ptbp1* cKO astrocytes clustered tightly together and no directional shift toward the neuronal cluster (***Figure 6—figure supplement 1 and –B***). Correlation analysis further supported this result, with a strong similarity between *Ptbp1* cKO and control astrocytes ($\rho$ =0.97), and low similarity between *Ptbp1* cKO astrocytes and astrocyte-derived neurons ($\rho$ =0.27) (***Figure 6—figure supplement 1C***). These findings indicate that, even with PTBP1 loss, cKO astrocytes retain a transcriptional profile very distinct from that of neurons, underscoring that *Ptbp1* deficiency alone does not induce astrocyte-to-neuron reprogramming at the transcriptomic level.

## Single-cell RNA-seq analysis reveals limited astrocyte-to-neuron conversion following genetic *Ptbp1* loss

Prior scRNA-seq study of *Ptbp1* null astrocytes was conducted only at 2 and 4 weeks after tamoxifen induction. To thoroughly assess the effects of *Ptbp1* loss in astrocytes at single-cell resolution, we performed scRNA-seq on equal proportions of tdTomato$^+$ (Cre+) and tdTomato$^-$ (Cre-) cells from the same animals 12 weeks after tamoxifen induction for the control and *Ptbp1* cKO groups (***Figure 7A***). By evaluating both Cre+ and Cre- populations in the cortices, we aimed to accurately identify cells with *Ptbp1* loss and assess potential cell-type conversion through clustering analysis. Following rigorous quality control and filtering, we obtained 10,851 cells from control samples and 8594 cells from *Ptbp1* cKO samples (***Figure 7B–D***). Based on gene expression profiles, cells were classified into ten distinct cell types: astrocytes (Astro), excitatory neurons (Exc), inhibitory neurons (Inh), microglia (Micro), immune cells (Immune), oligodendrocytes (OL), endothelial cells (Endo), pericytes (Peri), vascular leptomeningeal cells (VLMC), and ependymal cells (Ependymal) (***Figure 7B***). Control and *Ptbp1* cKO groups have almost identical UMAP distribution (***Figure 7C and D***). We confirmed the Cre transgene expression was predominantly limited to astrocytes (***Figure 7E***), ensuring that *Ptbp1* cKO was restricted to this cell type. Cell counts based on cell clustering revealed no significant differences in cell type proportions between control and *Ptbp1* cKO groups for both Cre +and Cre- cell populations (***Figure 7F and G***), suggesting that the loss of *Ptbp1* in astrocytes does not grossly affect cell type distribution in the cortex.

Although we observed no changes in cell type proportion, we identified a very small subset of excitatory neurons expressing the Cre transgene (***Figure 7F***). Specifically, there were two distinct excitatory neuron subpopulations, Exc-1 and Exc-2 (***Figure 7B***). Only the Exc-1 cluster contained Cre+ cells, all of which were exclusively from the *Ptbp1* cKO samples (***Figure 7H***). Out of the 30 cKO cells in the Exc-1 cluster, 7 were Cre+, suggesting a restricted and selective expression pattern in this subset.

However, of all the Cre+ cells from *Ptbp1* cKO samples (n=1460), only 0.48% (=7/1460) were categorized as excitatory neurons, while 90.62% (=1323/1460) were astrocytes (***Figure 7G***). This observation shows that the loss of PTBP1 does not effectively drive astrocyte-to-neuron conversion. The presence of these Cre+ Exc-1 cells, exclusively in *Ptbp1* cKO but not in control samples, hints at the potential of a very small subset of Aldh1l1+ astrocytes (<1%) for *Ptbp1* loss to facilitate neuron-like identity. On the other hand, we cannot exclude the possibility of rare neuronal expression of Aldh1l1-Cre/ERT2 in individual animals, since in our lineage tracing experiments we also observed a very small fraction (<1%) of tdT$^+$NeuN$^+$ cells in animals (***Figures 1H and 2G***).

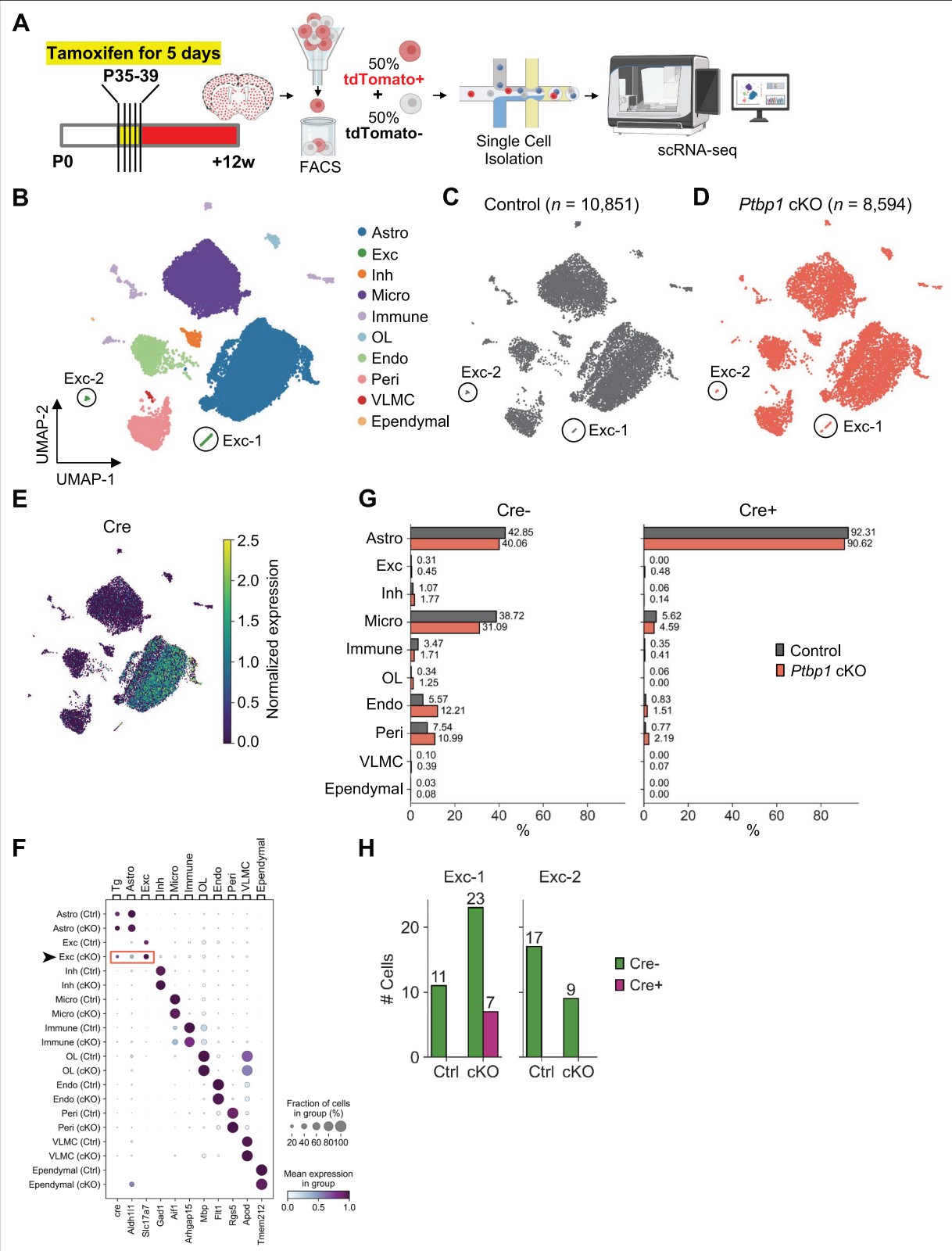

**Figure 7.** Single-cell RNA-seq analysis of *Ptbp1* conditional knockout (cKO) astrocytes shows limited astrocyte-to-neuron conversion. (**A**) Schematic of the experimental design of single-cell RNA-seq. (**B**) UMAP plot of all identified cell types based on gene expression profiles. Cells were classified into ten distinct cell types: astrocytes (Astro), excitatory neurons (Exc), inhibitory neurons (Inh), microglia (Micro), immune cells (Immune), oligodendrocytes (OL), endothelial cells (Endo), pericytes (Peri), vascular leptomeningeal cells (VLMC), and ependymal cells (Ependymal). Two excitatory neuron

*Figure 7 continued on next page*

*Figure 7 continued*

subpopulations, Exc-1 and Exc-2, are highlighted. (**C, D**) UMAP plots showing the distribution of cells in control (n=10,851) (**C**) and *Ptbp1* cKO (n=8594) (**D**) samples. (**E**) The Cre transgene expression projected on the UMAP plot. (**F**) Dot plot representing the expression of marker genes across identified cell types. (**G**) Bar plot showing the proportion of each cell type in control and *Ptbp1* cKO samples. (**H**) Bar plot showing the number of Cre-negative and Cre-positive cells in Exc-1 and Exc-2 clusters for control and *Ptbp1* cKO samples. The coronal section drawing in (**A**) was created using BioRender. com.

## Discussion

Using a genetic *Ptbp1* knockout approach in combination with lineage tracing and transcriptomic analyses, we found little evidence to support that *Ptbp1* loss of function efficiently induces astrocyte-to-neuron conversion in the adult mouse brain. Our lineage tracing data accord with recent studies that examined the effects of *Ptbp1* loss through AAV, Cas13X, ASO, and genetic *Ptbp1* knockout approaches (*Chen et al., 2022*; *Guo et al., 2022*; *Hoang et al., 2022*; *Hoang et al., 2023*; *Wang et al., 2021*; *Yang et al., 2023b*). Our RNA-seq data from *Ptbp1*-depleted astrocytes shows no enhancement of neuron-related gene expression, and our scRNA-seq analysis did not uncover any widespread changes in cell type proportion at 12 weeks after *Ptbp1* depletion apart from a very small subgroup of Cre+ neurons with unknown origin.

Our study is the first to examine RNA splicing changes in *Ptbp1* cKO astrocytes. In contrast to minimal gene expression changes, *Ptbp1* cKO astrocytes exhibit widespread splicing alterations, confirming PTBP1's splicing regulatory function and a clear loss of PTBP1 function in our isolated astrocytes. As a well-characterized repressor of neuronal RNA splicing, PTBP1 has been shown to regulate brain-specific exons (*Boutz et al., 2007*; *Llorian et al., 2010*; *Makeyev et al., 2007*). Our analysis in mature astrocytes suggests that PTBP1 regulates a core astrocytic splicing network distinct from its neuronal or developmental splicing regulatory role. While the function of astrocyte-specific PTBP1 splicing regulation remains to be understood, our data indicate that loss of *Ptbp1* in mature astrocytes is not sufficient to induce neuron-specific splicing patterns, unlike its well-established effect in other non-neuronal cell types.

Previous genetic *Ptbp1* knockout studies have used mutant mice harboring loxP sites flanking the *Ptbp1* promoter and the first exon (*Hoang et al., 2022*; *Shibayama et al., 2009*). Our study produced similar findings by using a different *Ptbp1*$^{loxP/loxP}$ mouse line carrying loxP sites that flank the second exon. We confirmed our *Ptbp1* depletion was exclusive and specific to astrocytes using the Aldh1l1-Cre/ERT2 transgenic mice and the Ai14 Cre-reporter mice (*Srinivasan et al., 2016*). Thus, our genetic targeting approach mitigates concerns of leaky neuronal Cre expression associated with GFAP-based reagents or promoters raised by previous studies (*Bocchi et al., 2022*; *Hoang et al., 2022*; *Hoang et al., 2023*; *Wang et al., 2021*). Our findings reinforce the conclusion that *Ptbp1* depletion in astrocytes is not sufficient to induce neuronal conversion irrespective of the genetic targeting method employed to induce *Ptbp1* loss. We did not detect broad induction of neuronal cell types among *Ptbp1* cKO astrocytes in our scRNA-seq analysis, but uncovered a very small subpopulation of Cre+ excitatory neurons among *Ptbp1* cKO cells. These cells make up <0.5% *Ptbp1*-depleted cells, indicating that while *Ptbp1* loss could promote acquisition of neuronal identity, such a phenomenon is exceedingly rare or may be an experimental artifact.

Discrepancies in previous reports on *Ptbp1* knockdown-mediated astrocyte-to-neuron conversion have raised arguments for phenotypic improvements and the cellular origin of converted neurons. GFAP promoter leakage in neurons may account for widespread reporter expression in endogenous neurons during AAV-based *Ptbp1* knockdown (*Wang et al., 2021*). Rescue of disease and aging phenotypes upon *Ptbp1* depletion in astrocytes in the absence of neuronal conversion has been speculated to occur by trophic remodeling of local cellular environment and suppression of pro-inflammatory responses in glia (*Fu and Mobley, 2023*). While it is possible that induction of the astrocyte-specific network of PTBP1-regulated splicing observed in *Ptbp1* cKO astrocytes could mediate a protective response supporting phenotypic recovery, our GO enrichment analysis did not uncover upregulation of ontologies related to neurotrophic effects or inflammatory responses. The possibility remains that other glial cell types are responsible for the increased neuronal populations reported in previous studies of *Ptbp1* knockdown. ASO-mediated knockdown of *Ptbp1* has been shown to activate neurogenesis in ependymal cells within neurogenic niches of the subventricular zone (SVZ) in aged mice (*Maimon and Chillon-Marinas, 2024*). Radial glial cells residing in the subgranular zone of the dentate

gyrus have also been shown to possess neurogenic capacity in adult rodents in physiological conditions and upon *Ptbp1* depletion (*Grelat et al., 2018*; *Maimon and Chillon-Marinas, 2024*). These glial-like subpopulations may represent the original sources of converted neurons, but the extent to which they obtain intrinsic neurogenic potential to be susceptible to reprogramming remains to be further investigated. More importantly, considering the stem-cell-like features of these cell populations, whether the observed reporter-positive new-born neurons originate from the reprogramming or reactivated proliferation still needs to be carefully characterized.

### Limitations of current studies and future directions

Molecular markers and transcriptomic analyses provide an important assessment of key molecular changes and alterations of transcriptome-defined cell states associated with *in vivo* reprogramming but do not capture any cellular or morphological transformation. Continuous live imaging of converted cells is necessary to document loss of glial morphology and acquisition of neuronal cytoarchitecture during the conversion process, which has only recently been performed *in vivo* (*Xiang et al., 2024*). While our data indicate that *Ptbp1* depletion in adult astrocytes is not sufficient to induce efficient neuronal conversion, targeting other cell types for *Ptbp1* loss of function may be possible to promote neurogenesis and rescue neurodegenerative dysfunction (*Böck et al., 2024*). The use of genetic knockout approaches, stringent lineage tracing analyses, and comprehensive transcriptomic and splicing profiling remains critical for future studies of cellular reprogramming.

# Materials and methods

**Key resources table**

| Reagent type (species) or resource | Designation | Source or reference | Identifiers | Additional information |
|---|---|---|---|---|
| Strain, strain background (*Mus musculus,* male, female) | "*Ptbp1*^loxP/loxP^;tdT^+/-^; Aldh1l1-Cre^+/-^" (*Ptbp1* cKO) | This paper | | See 'Mouse maintenance' |
| Antibody | Anti-S100β (mouse monoclonal) | Sigma-Aldrich | Cat#: S2532; RRID:AB_477499 | IF(1:500) |
| Antibody | Anti-PTBP1 (rabbit polyclonal) | Gift from Dr. Douglas Black; *Markovtsov et al., 2000* | | IF(1:1000) |
| Antibody | Anti-PTBP2 (rabbit polyclonal) | Gift from Dr. Douglas Black; *Sharma et al., 2005* | | IF(1:1000) |
| Antibody | Anti-NeuN (mouse monoclonal) | EMD Millipore | Cat#:MAB377;RRID:AB_2298772 | IF(1:400) |
| Antibody | Anti-Iba1 (goat polyclonal) | Abcam | Cat#:ab5076; RRID:AB_2224402 | IF(1:1000) |
| Antibody | Anti-tyrosine hydroxylase (chicken polyclonal) | Aves Labs | Cat#: TYH; RRID:AB_10013440 | IF(1:1000) |
| Antibody | Anti-GFAP (mouse monoclonal) | Cell Signaling Technologies | Cat#: 3670; RRID:AB_561049 | IF(1:500) |
| Sequence-based reagent | PTBP1 | This paper | PCR primers | Forward: TTGCCTCCTTTGAGCAACTT Reverse: TTTGCGACATTTCTCTGCAC |
| Sequence-based reagent | Aldh1l1-Cre/ERT2 | This paper | PCR primers | Forward: CTTCAACAG GTGC CTTCCA Reverse: GGCAAACGG ACAGAAGCA |
| Sequence-based reagent | Ai14—WT | This paper | PCR primers | Forward: AAG GGA GCT GCAG TGGAGTA Reverse: CCG AAA ATC TGTG GGAAGTC |

*Continued on next page*

*Continued*

| Reagent type (species) or resource | Designation | Source or reference | Identifiers | Additional information |
|---|---|---|---|---|
| Sequence-based reagent | Ai14—mutant | This paper | PCR primers | Forward: CTGTTCCTG TACGGCATG G<br>Reverse: GGCATTAAA GCAG CGTATCC |
| Other | DAPI stain | Sigma-Aldrich | D9542 | 2 µg/ml |

## Mouse maintenance

Mice in this study were maintained, and the related experimental protocols were used in compliance with the requirements of the Institutional Animal Care and Use Committees (IACUC) at the University of California, Riverside. All mice were housed with a temperature at 22 ± 2°C and a 12-hour light/dark cycle under the monitoring of the veterinary and staff. Both males and females were used in this study. The three founder mouse lines are as below:

1. *Ptbp1*$^{loxP/loxP}$ line carries loxP sites flanking *Ptbp1* exon 2 was provided by Dr. Douglas Black (*Stork et al., 2019*; *Yeom et al., 2018*).
2. The tamoxifen-inducible astrocyte-specific Cre line, Tg (Aldh1l1-Cre/ERT2), was provided by Dr. Todd Fiacco (https://www.jax.org/strain/029655).
3. The Cre-dependent tdTomato reporter line Ai14 (LSL-tdTomato-WPRE) was provided by Dr. Martin M. Riccomagno (https://www.jax.org/strain/007914#).

## Mouse genotyping

Mouse genotyping was performed by extracting DNA from mouse toe or tail biopsy samples by incubating samples in 100 µl lysis solution (Bioland, #GT0102) at 100°C for 30 minutes, after which the reaction was neutralized by adding 100 µl DNA stabilization solution (Bioland, #GT0102). Genotyping was performed by PCR (94°C for 4 minutes, followed by 34 cycles of amplification: 94°C for 30 seconds, 55°C for 30 seconds, 72°C for 30 seconds, then 72°C for 5 minutes, and hold at 4°C) with the listed primer sets.

| Primer name | Species | Sequence—forward | Sequence—reverse |
|---|---|---|---|
| PTBP1 | Mouse | TTG CCT CCT TTG AGC AAC TT | TTT GCG ACA TTT CTC TGC AC |
| Aldh1l1-Cre/ERT2 | Mouse | CTT CAA CAG GTG CCT TCC A | GGC AAA CGG ACA GAA GCA |
| Ai14—WT | Mouse | AAG GGA GCT GCA GTG GAG TA | CCG AAA ATC TGT GGG AAG TC |
| Ai14—mutant | Mouse | CTG TTC CTG TAC GGC ATG G | GGC ATT AAA GCA GCG TAT CC |

## Tamoxifen administration

Tamoxifen powder (Sigma #T5648) was dissolved in corn oil (Sigma #C8267) to 10 mg/ml, kept in 4°C in dark up to 1 week. To induce astrocyte-specific Cre expression in adult mouse brain, five consecutive doses of tamoxifen were administered by IP injection as 75 mg/kg to mice from P35 to 39.

## Immunostaining

At the desired time points, mice were anesthetized by $CO_2$, perfused with cold phosphate-buffered saline (PBS) (pH 7.4) and cold 4% paraformaldehyde (PFA) (Acros Organics #AC416785000) in PBS. Mouse brains were dissected out and post-fixed in 4% PFA at 4°C overnight. On the following day, the brains were washed three times with pH 7.4 PBS, embedded in 3% agarose in PBS, and then sectioned to 100 µm using a Vibratome LEICA VT1000S (Leica). Immunostaining was carried out as previously described. Briefly, brain sections were rinsed three times (30 minutes/each) with PBS, permeabilized with 0.5% Triton X-100 in PBS for 30 minutes, incubated in a blocking buffer (10% donkey serum, 2% bovine serum albumin [BSA], 0.3% Triton X-100 in pH7.4 PBS) for 1 hour at room temperature (RT). Then the sections were incubated with appropriate primary antibodies in blocking buffer at 4°C overnight. The primary antibodies used in this study are mouse S100β (Sigma #S2532, 1:500), rabbit PTBP1 (1:1000, gift from Dr. Douglas Black , *Markovtsov et al., 2000*), rabbit PTBP2 (1:1000, gift from Dr.

Black and from this study, *Sharma et al., 2005*), mouse NeuN (EMD Millipore # MAB377, 1:400), goat Iba1 (Abcam #ab5076, 1:1000), chicken tyrosine hydroxylase (TH, Aves labs #TYH, 1:1000), and mouse GFAP (Cell Signaling Technology #3670, 1:500). On the second day, sections were rinsed three times with 0.3% Triton X-100 in PBS and incubated in appropriate Alexa Fluor secondary antibodies (Life Technologies, 1:1000) in blocking buffer at 4°C overnight. On the third day, the sections were washed three times with PBS, incubated with DAPI in PBS (Sigma #D9542, 1:500) for 1 hour at RT, washed three times again with PBS, and then mounted with ProLong Gold Antifade Mountant (Thermo Fisher Scientific # P36930). Mounted sections were left in the dark at RT overnight and imaged with confocal microscopy LSM800 (Zeiss).

## Imaging and quantification

Single-focal plan or z-stack images were obtained from mouse brain sections by LSM800 (Zeiss) and Zen blue software. Image processing and automated cell counting for DAPI and NeuN were done using Zen blue software. The cell counting for tdT$^+$, PTBP1$^+$ cells was done manually. All data analyses were carried out using Excel. The significance was evaluated by *t*-test in Excel. $p<0.05$ was considered significant. *$p<0.05$, **$p<0.01$ and ***$p<0.001$.

## Single-cell isolation

To obtain the single-cell suspension from adult mouse brain, we adapted the isolation method from a previous publication (*Swartzlander et al., 2018*) and then further purified the samples with the Debris Removal reagent (Miltenyi Biotec #130-109-398) according to vendor's manual. To inhibit RNase activity, we supplemented the cell dissociation buffer and resuspension buffer with RNase inhibitor (Sigma #3335402001).

The mouse was perfused with 100 ml cold PBS to thoroughly wash out blood. The desired brain regions were dissected out, rinsed with cold Hank's Balanced Salt Solution (HBSS), gently chopped into 1–2 mm, and incubated in 2.5 ml HBSS with 20 U/ml papain (Worthington Biochemical #LK003176) and 0.005% DNase I (Sigma #11284932001) in 37°C for 15 minutes with gentle agitation every 5 minutes. The tissue lysis was triturated four times using a silanized glass pipet (Thermo Fisher Scientific #NC0319875) and then another 15-minute incubation at 37°C as above. After the completion of the lysis, 2.5 ml ice-cold HBSS+ (HBSS with 0.2 U/ul RNase inhibitor, 0.5% BSA, and 2 mM ethylenediaminetetraacetic acid [EDTA]) was added, and the sample was centrifuged at $300 \times g$ at 4°C for 5 minutes. Then the supernatant was removed, and 1 ml ice-cold HBSS+ was added to the cell pellets. The sample was triturated three times using fire-polished silanized glass pipet, transferred into a pre-chilled 15 ml tube, and centrifuged at $100 \times g$ at 4°C for 15 seconds. The supernatant at step was considered a single-cell suspension and transferred into a clean 15 ml tube on ice. A new 1 ml ice-cold HBSS+ was added to the cell pellets, the above steps were repeated 3–4 times, and the single-cell suspension was pooled together (around 3–4 ml). The single-cell suspension was filtered by 30 µm MACS SmartStrainer (Miltenyi Biotec #130098458) to remove cell debris and large clumps and then was subjected to the Debris Removal reagent for further purification following the vendor's manual. The single cells were resuspended in 500 ul cold cell resuspension buffer (1× PBS with 0.5% BSA and 1.0 U/µl RNase Inhibitor) and followed by tdT$^+$ cell sorting.

## Fluorescence-activated cell sorting (FACS)

Cells were sorted by using MoFlo Astrios EQ Cell Sorter (Beckman Coulter) with Summit version 6.3 software. tdTomato was excited by a 561 nm laser and detected by emissions spectra between 561–579 nm. Total cell population was selected by forward and side scatter profiles, and doublet populations were excluded by side scatter area vs. side scatter height plot gating only singlets. Fluorescence gates were established by comparing tdT$^+$ and tdT$^-$ cells, with tdT$^+$ cells sorted for RNA extraction into cold Trizol LS. The FAC sorting process was completed within 2–3 hours per sample.

## Bulk RNA-sequencing

For bulk RNA-sequencing, mice at 4 weeks after tamoxifen injection were collected for single tdT$^+$ cell sorting. Differently, the single cells were directly sorted into a 1.5 ml Eppendorf tube containing 750 µl of Trizol LS (Thermo Fisher Scientific #10296028). Cell sorting in each tube proceeded until volume reached 1 ml and then a new tube with Trizol LS was replaced until the end of the sorting. Total RNA

was extracted according to the manufacturer's instructions for Trizol reagent. The RNA quality was monitored by Agilent TapeStation 4150 in the Genomics core facility in UCR. The samples that passed the quality check were sent to Novogene for library preparation and sequencing.

For quality control and alignment of raw bulk RNA-seq reads from *Ptbp1* knockout samples, we employed the Snakemake workflow 'preprocessing_RNAseq.smk' from SnakeNgs (version 0.2.0; https://github.com/NaotoKubota/SnakeNgs; *Kubota, 2025*). In brief, low-quality reads were filtered using fastp (version 0.23.4; *Chen et al., 2018*), and the remaining reads were aligned to the mouse reference genome (mm10) using STAR (version 2.7.11a; *Dobin et al., 2013*), with Ensembl transcript annotation (v102). Quality metrics were then collected with Picard (version 3.1.1; http://broadinstitute.github.io/picard/) and summarized using MultiQC (version 1.25; *Ewels et al., 2016*). We applied Shiba (version 0.4.0; *Kubota et al., 2024*) to the mapped reads with the default parameters for differential gene expression and splicing analysis. Shiba internally employs DESeq2 (*Love et al., 2014*) for differential gene expression analysis. Genes with adjusted p<0.05 and the absolute log2 fold change >1 were considered DEGs. For differential splicing analysis, events with adjusted Fisher's exact test p<0.05, dPSI >10, and Welch's *t*-test p<0.05 were considered DSEs. Enrichment analysis was performed for DSGs using Metascape (*Zhou et al., 2019*).

Motif enrichment analysis was performed using XSTREME from the MEME Suite (version 5.5.7; *Bailey et al., 2015*). Enrichment ratios and associated *q*-values for each identified motif were calculated using the Simple Enrichment Analysis (SEA) algorithm implemented in XSTREME. For this analysis, 200 bp intronic sequences immediately upstream and downstream of differentially activated (upregulated) and repressed (downregulated) skipped exons were extracted and used as target sequences. As background controls, we used the corresponding upstream and downstream regions from all expressed skipped exons.

For comparison of the transcriptome signatures with our *Ptbp1* knockout samples, we analyzed RNA-seq data from mouse cortex across various developmental stages and ages (*Weyn-Vanhentenryck et al., 2018*), as well as from *in vitro* differentiated neurons (DIV-8 [embryonic stem cells], DIV-4 [neuroepithelial stem cells], DIV 0 [radial glia], DIV-1, DIV-7, DIV-16, DIV-21, and DIV-28) (*Hubbard et al., 2013*) and astrocyte-derived neurons generated through overexpression of proneural transcription factors (Ngn2 or mutant PmutNgn2) (*Pereira et al., 2024*). We employed the Snakemake workflow 'preprocessing_RNAseq.smk' for quality control and read alignment. Shiba was used to perform differential gene expression and splicing analysis and quantify transcript per million (TPM) for each gene and PSI for each alternative splicing event. Correlations in gene expression and splicing profiles were then assessed for DEGs and DSEs detected either in astrocytes or neuronal samples.

## Single-cell RNA sequencing

Both mouse brain cortices were collected 12 weeks after tamoxifen injection and subjected to single-cell isolation. After cell sorting, the single-cell suspension of tdT+ cells was approximately adjusted to 1000 cells/ul, and the procedures were carried out according to the manufacturer's instructions for Chromium Next GEM Single Cell 3' GEM, Library & Gel Bead Kit v3.1 (10x Genomics # 1000128). To enrich the tdT+ cells, for each animal, the cells subjected to 10× procedure were mixed as a 1:1 ratio of tdT+ and tdT- from the same animal.

For quantification of unique molecular identifiers (UMIs) in our scRNA-seq reads from *Ptbp1* knockout samples, we employed the Snakemake workflow 'kb-nac.smk' from SnakeNgs (version 0.2.0). This workflow internally executes kb-python (*Sullivan et al., 2024*), a wrapper of kallisto and bustools (*Melsted et al., 2021*), to obtain UMI count matrix. We ran this workflow with customized genome and transcript annotation files that contain tdTomato and Cre gene sequences as well as the reference Gencode mouse genes. The UMI count matrix was then processed by Scanpy (version 1.9.5; *Wolf et al., 2018*). We used Scrublet (*Wolock et al., 2019*) for cell doublets detection and Harmony (*Korsunsky et al., 2019*) for batch-effect correction. Cell types were manually assigned based on the expression level of marker genes.

## Materials availability statement

Newly generated materials are available from the corresponding author upon reasonable request and completion of a standard material transfer agreement.

## Acknowledgements

This work was supported by the NIH Research Project Grant R01NS125276 (SZ). No previously published material has been adapted or reproduced in this article.

## Additional information

### Funding

| Funder | Grant reference number | Author |
| --- | --- | --- |
| National Institute of Health | R01NS125276 | Sika Zheng |

The funders had no role in study design, data collection and interpretation, or the decision to submit the work for publication.

### Author contributions

Min Zhang, Formal analysis, Investigation, Visualization, Methodology, Writing – original draft, Writing – review and editing; Naoto Kubota, Data curation, Software, Formal analysis, Validation, Investigation, Writing – original draft, Writing – review and editing; David Nikom, Validation, Investigation, Visualization, Writing – original draft, Writing – review and editing; Ayden Arient, Investigation; Sika Zheng, Conceptualization, Resources, Supervision, Funding acquisition, Methodology, Project administration, Writing – review and editing

### Author ORCIDs

Min Zhang ⓘD https://orcid.org/0009-0008-4105-057X
Naoto Kubota ⓘD https://orcid.org/0000-0003-0612-2300
David Nikom ⓘD https://orcid.org/0000-0003-2722-6002
Sika Zheng ⓘD https://orcid.org/0000-0002-0573-4981

### Ethics

All animal procedures were approved by the University of California, Riverside Institutional Animal Care and Use Committee (IACUC: #111), in accordance with the guidelines of the US Department of Agriculture, the International Association for the Assessment and Accreditation of Laboratory Animal Care, and the National Institutes of Health.

Reviewer #1 (Public review): https://doi.org/10.7554/eLife.107683.3.sa1
Reviewer #2 (Public review): https://doi.org/10.7554/eLife.107683.3.sa2
Author response https://doi.org/10.7554/eLife.107683.3.sa3

## Additional files

### Supplementary files

MDAR checklist

### Data availability

Raw fastq and processed files of bulk RNA-seq and scRNA-seq data have been deposited at Gene Expression Omnibus under accession GSE294763 and GSE294768. The code used for analyzing the sequencing data have been deposited at GitHub (https://github.com/Sika-Zheng-Lab/Ptbp1_astrocyte_2025 copy archived at *Zhang et al., 2025*). All data generated or analyzed during this study are included in the manuscript and supporting files; source data files have been provided.

The following datasets were generated:

| Author(s) | Year | Dataset title | Dataset URL | Database and Identifier |
|---|---|---|---|---|
| Zhang M, Kubota N, Nikom D, Arient A, Zheng S | 2025 | Genetic Ptbp1 depletion does not induce neuronal RNA splicing patterns in mature astrocytes (Bulk RNA-seq) | https://www.ncbi.nlm.nih.gov/geo/query/acc.cgi?acc=GSE294763 | NCBI Gene Expression Omnibus, GSE294763 |
| Zhang M, Kubota N, Nikom D, Arient A, Zheng S | 2025 | Genetic Ptbp1 depletion does not induce neuronal RNA splicing patterns in mature astrocytes (Single-cell RNA-seq) | https://www.ncbi.nlm.nih.gov/geo/query/acc.cgi?acc=GSE294768 | NCBI Gene Expression Omnibus, GSE294768 |

The following previously published dataset was used:

| Author(s) | Year | Dataset title | Dataset URL | Database and Identifier |
|---|---|---|---|---|
| Pereira A, Diwakar J, Masserdotti G, Beşkardeş S, Simon T, So Y, Martín-Loarte L, Bergemann F, Vasan L, Schauer T, Danese A, Bocchi R, Colomé-Tatché M, Schuurmans C, Philpott A, Straub T, Bonev B, Götz M | 2024 | Direct neuronal reprogramming of mouse astrocytes is associated with multiscale epigenome remodeling and requires Yy1 | https://www.ncbi.nlm.nih.gov/geo/query/acc.cgi?acc=GSE208742 | NCBI Gene Expression Omnibus, GSE208742 |

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
