## [Editor Report · eLife Assessment]

This study reports **important** negative results, showing that genetically removing the RNA-binding protein PTBP1 in astrocytes is insufficient to convert them into neurons, thereby challenging previous claims in the field. It also offers a **compelling** analysis of PTBP1's role in regulating astrocyte-specific splicing. The evidence is strong, as the experiments are technically sound, carefully controlled, and supported by both imaging and transcriptomic analyses.

---

## [Referee Report · Reviewer #1 (Public review)]

Summary:

Zhang et al. used a conditional knockout mouse model to re-examine the role of the RNA-binding protein PTBP1 in the transdifferentiation of astroglial cells into neurons. Several earlier studies reported that PTBP1 knockdown can efficiently induce the transdifferentiation of rodent glial cells into neurons, suggesting potential therapeutic applications for neurodegenerative diseases. However, these findings have been contested by subsequent studies, which in turn have been challenged by more recent publications. In their current work, Zhang et al. deleted exon 2 of the Ptbp1 gene using an astrocyte-specific, tamoxifen-inducible Cre line and investigated - using fluorescence imaging and bulk and single-cell RNA-sequencing - whether this manipulation promotes the transdifferentiation of astrocytes into neurons across various brain regions. The data strongly indicate that genetic ablation of PTBP1 is not sufficient to drive efficient conversion of astrocytes into neurons. Interestingly, while PTBP1 loss alters splicing patterns in numerous genes, these changes do not shift the astroglial transcriptome toward a neuronal profile.

Strengths:

Although this is not the first report of PTBP1 ablation in mouse astrocytes *in vivo*, this study utilizes a distinct knockout strategy and provides novel insights into PTBP1-regulated splicing events in astrocytes. The manuscript is well written, and the experiments are technically sound and properly controlled. I believe this study will be of considerable interest to the broad readership of eLife.

Original weaknesses:

(1) The primary point that needs to be addressed is a better understanding of the effect of exon 2 deletion on PTBP1 expression. Figure 4D shows successful deletion of exon 2 in knockout astrocytes. However - assuming that the coverage plots are CPM-normalized - the overall PTBP1 mRNA expression level appears unchanged. Figure 6A further supports this observation. This is surprising, as one would expect that the loss of exon 2 would shift the open reading frame and trigger nonsense-mediated decay of the PTBP1 transcript. Given this uncertainty, the authors should confirm the successful elimination of PTBP1 protein in cKO astrocytes using an orthogonal approach, such as Western blotting, in addition to immunofluorescence. They should also discuss possible reasons why PTBP1 mRNA abundance is not detectably affected by the frameshift.

(2) The authors should analyze PTBP1 expression in WT and cKO substantia nigra samples shown in Figure 3 or justify why this analysis is not necessary.

(3) Lines 236-238 and Figure 4E: The authors report an enrichment of CU-rich sequences near PTBP1-regulated exons. To better compare this with previous studies on position-specific splicing regulation by PTBP1, it would be helpful to assess whether the position of such motifs differs between PTBP1-activated and PTBP1-repressed exons.

(4) The analyses in Figure 5 and its supplement strongly suggest that the splicing changes in PTBP1-depleted astrocytes are distinct from those occurring during neuronal differentiation. However, the authors should ensure that these comparisons are not confounded by transcriptome-wide differences in gene expression levels between astrocytes and developing neurons. One way to address this concern would be to compare the new PTBP1 cKO data with publicly available RNA-seq datasets of astrocytes induced to transdifferentiate into neurons using proneural transcription factors (e.g., PMID: 38956165).

Point 1 has been successfully addressed in the revision by providing relevant references/discussion. Points 2-4 were addressed by including additional data/analyses.

---

## [Referee Report · Reviewer #2 (Public review)]

Summary:

The manuscript by Zhang and colleagues describes a study that investigated if deletion of PTBP1 in adult astrocytes in mice led to an astrocyte-to-neuron conversion. The study revisited the hypothesis that reduced PTBP1 expression reprogrammed astrocytes to neurons. More than 10 studies have been published on this subject, with contradicting results. Half of the studies supported the hypothesis while the other half did not. The question being addressed is an important one because if the hypothesis is correct, it can lead to exciting therapeutic applications for treating neurodegenerative diseases such as Parkinson's disease.

In this study, Zhang and colleagues conducted a conditional mouse knockout study to address the question. They used the Cre-LoxP system to specifically delete PTBP1 in adult astrocytes. Through a series of carefully controlled experiments including cell lineage tracing, the authors found no evidence for the astrocyte-to-neuron conversion.

The authors then carried out a key experiment that none of previous studies on the subject did: investigating alternative splicing pattern changes in PTBP1-depleted cells using RNA-seq analysis. The idea is to compare the splicing pattern change caused by PTBP1 deletion in astrocytes to what occurs during neurodevelopment. This is an important experiment that will help illuminate if the astrocyte-to-neuron transition occurred in the system. The result was consistent with that of the cell staining experiments: no significant transition being detected.

These experiments demonstrate that, in this experiment setting, PTBT1 deletion in adult astrocytes did not convert the cells to neurons.

Strengths:

This is a well-designed, elegantly conducted, and clearly described study that addresses an important question. The conclusions provide important information to the field.

To this reviewer, this study provided convincing and solid experimental evidence to support the authors' conclusions.

My concerns in the previous review have been addressed satisfactorily.

---

## [Author Response]

The following is the authors’ response to the original reviews.

**Reviewer #1 (Public review):**
Summary:Zhang et al. used a conditional knockout mouse model to re-examine the role of the RNAbinding protein PTBP1 in the transdifferentiation of astroglial cells into neurons. Several earlier studies reported that PTBP1 knockdown can efficiently induce the transdifferentiation of rodent glial cells into neurons, suggesting potential therapeutic applications for neurodegenerative diseases. However, these findings have been contested by subsequent studies, which in turn have been challenged by more recent publications. In their current work, Zhang et al. deleted exon 2 of the Ptbp1 gene using an astrocyte-specific, tamoxifen-inducible Cre line and investigated, using fluorescence imaging and bulk and single-cell RNA-sequencing, whether this manipulation promotes the transdifferentiation of astrocytes into neurons across various brain regions. The data strongly indicate that genetic ablation of PTBP1 is not sufficient to drive efficient conversion of astrocytes into neurons. Interestingly, while PTBP1 loss alters splicing patterns in numerous genes, these changes do not shift the astroglial transcriptome toward a neuronal profile.Strengths:Although this is not the first report of PTBP1 ablation in mouse astrocytes *in vivo*, this study utilizes a distinct knockout strategy and provides novel insights into PTBP1-regulated splicing events in astrocytes. The manuscript is well written, and the experiments are technically sound and properly controlled. I believe this study will be of considerable interest to a broad readership.Weaknesses:(1) The primary point that needs to be addressed is a better understanding of the effect of exon 2 deletion on PTBP1 expression. Figure 4D shows successful deletion of exon 2 in knockout astrocytes. However, assuming that the coverage plots are CPM-normalized, the overall PTBP1 mRNA expression level appears unchanged. Figure 6A further supports this observation. This is surprising, as one would expect that the loss of exon 2 would shift the open reading frame and trigger nonsense-mediated decay of the PTBP1 transcript. Given this uncertainty, the authors should confirm the successful elimination of PTBP1 protein in cKO astrocytes using an orthogonal approach, such as Western blotting, in addition to immunofluorescence. They should also discuss possible reasons why PTBP1 mRNA abundance is not detectably affected by the frameshift.

We thank the reviewer for raising this important point. Indeed, the deletion of exon 2 introduces a frameshift that is predicted to disrupt the PTBP1 open reading frame and trigger nonsensemediated decay (NMD). While our CPM-normalized coverage plots (Figure 4D) and gene-level expression analysis (Figure 6A) suggest that PTBP1 mRNA levels remain largely unchanged in cKO astrocytes, we acknowledge that this observation is counterintuitive and merits further clarification.

We suspect that the process of brain tissue dissociation and FACS sorting for bulk or single cell RNA-seq may enrich for nucleic material and thus dilute the NMD signal, which occurs in the cytoplasm. Alternatively, the transcripts (like other genes) may escape NMD for unknown mechanisms. Although a frameshift is a strong indicator for triggering NMD, it does not guarantee NMD will occur in every case. (lines 346-353)

Regarding the validation of PTBP1 protein depletion in cKO astrocytes by Western blotting, we acknowledge that orthogonal approaches to confirm PTBP1 elimination would address uncertainty around the effect of exon 2 deletion on PTBP1 expression. The low cell yield of cKO astrocytes vis FACS poses a significant burden on obtaining sufficient samples for immunoblotting detection of PTBP1 depletion. On average 3-5 adult animals per genotype (with three different alleles) are needed for each biological replicate. The manuscript contains PTBP1 immunofluorescence staining of brain slides to demonstrate PTBP1 deletion (Figures 1-2, Figure 3 supplement 1). Our characterization of this Ptbp1 deletion allele in other contexts show the loss of full length PTBP1 proteins in ESCs using Western blotting (PMID: 30496473). Furthermore, germline homozygous mutant mice do not survive beyond embryonic day 6, supporting that it is a loss of function allele.

(2) The authors should analyze PTBP1 expression in WT and cKO substantia nigra samples shown in Figure 3 or justify why this analysis is not necessary.

We thank the reviewer for pointing out this important question. Although we are using an astrocyte-specific PTBP1 knockout (KO) mouse model, which is designed to delete PTBP1 in all the astrocyte throughout mouse brain, and although we have systematically verified PTBP1 elimination in different mouse brain regions (cortex and striatum) at multiple time points (from 4w to 12w after tamoxifen administration), we agree that it remains necessary and important to demonstrate whether the observed lack of astrocyte-to-neuron conversion is indeed associated with sufficient PTBP1 depletion.

We have analyzed the PTBP1 expression in the substantia nigra, as we did in the cortex and striatum. We added a new figure (Figure 3-figure supplement 1) to show the results. We found in cKO samples, tdT+ cells lack PTBP1 immunostaining, and there is no overlapping of NeuN+ and tdT+ signals. These results show effective PTBP1 depletion in the substantia nigra, similar to that observed in the cortex and striatum. (line 221-224)

(3) Lines 236-238 and Figure 4E: The authors report an enrichment of CU-rich sequences near PTBP1-regulated exons. To better compare this with previous studies on position-specific splicing regulation by PTBP1, it would be helpful to assess whether the position of such motifs differs between PTBP1-activated and PTBP1-repressed exons.

We thank the reviewer for this insightful comment. We agree that assessing the positional distribution of CU-rich motifs between PTBP1-activated and PTBP1-repressed exons would provide valuable insight into the position-specific regulatory mechanisms of PTBP1. In response, we have performed separate motif enrichment analyses for PTBP1-activated and PTBP1-repressed exons and examined whether their positional patterns differ (Figure 4–figure supplement 2).

Our analysis revealed that CU-rich motifs were significantly enriched in the upstream introns of both activated and repressed exons by PTBP1 loss, with higher enrichment observed in repressed exons (Enrichment ratio = 2.14, q = 9.00×10-5) compared to activated exons (Enrichment ratio = 1.72, q = 7.75×10-5) (Figure 4–figure supplement 2B–C). In contrast, no CU-rich motifs were found downstream of activated exons (Figure 4–figure supplement 2D), while a weak, non-significant enrichment was observed downstream of repressed exons (Enrichment ratio = 1.21, q = 0.225; Figure 4–figure supplement 2E). These results do not necessarily fully fit with a couple of earlier PTBP1 CLIP studies showing differential PTBP1 binding for repressed vs activated exons but are more in line with the Black Lab study (PMID: 24499931) that PTBP1 binds upstream introns of both repressed and activated exons. Either case, PTBP1 affects a diverse set of alternative exons and likely involves diverse contextdependent binding patterns (lines 244-257).

(4) The analyses in Figure 5 and its supplement strongly suggest that the splicing changes in PTBP1-depleted astrocytes are distinct from those occurring during neuronal differentiation. However, the authors should ensure that these comparisons are not confounded by transcriptome-wide differences in gene expression levels between astrocytes and developing neurons. One way to address this concern would be to compare the new PTBP1 cKO data with publicly available RNA-seq datasets of astrocytes induced to transdifferentiate into neurons using proneural transcription factors (e.g., PMID: 38956165).

We would like to express our gratitude for the thoughtful feedback. We agree that transcriptome-wide differences in gene expression between astrocytes and developing neurons could confound the interpretation of splicing differences. To address this concern, we have incorporated publicly available RNA-seq datasets from studies in which astrocytes are reprogrammed into neurons using proneural transcription factors, Ngn2 or PmutNgn2 (PMID: 38956165).

The results of principal component analysis (PCA) for splicing profiles revealed that the *in vivo* splicing profiles from this study and the *in vitro* splicing profiles from PMID 38956165 are well separated on PC1 and PC2. While Ngn2/PmutNgn2-induced neurons and control astrocytes started to show distinction on PC3 (and to some degree on PC4), Ptbp1 cKO samples remained tightly grouped with control astrocytes and showed no directional shift toward the neuronal cluster (Figure 5–figure supplement 2B). These findings further support the conclusion that PTBP1 depletion in mature astrocytes does not induce a neuronal-like splicing program, even when compared against neurons derived from the astrocyte lineage (lines 306318).

The pairwise correlation analysis of percent spliced in between Ptbp1 cKO, control astrocytes, and induced neurons confirmed that Ptbp1 cKO astrocytes are highly similar to control astrocytes (ρ = 0.81) and clearly distinct from induced neurons (ρ = 0.62) (Figure 5– figure supplement 2C), reinforcing the notion that PTBP1 loss alone is insufficient to drive a neuronal-like splicing transition (lines 319-336).

Consistent with the analysis for splicing profiles, PCA for gene expression profiles showed that control and Ptbp1 cKO astrocytes clustered tightly together and no directional shift toward the neuronal cluster while Ngn2/PmutNgn2-induced neurons and control astrocytes were distributed across a broader range (Figure 6–figure supplement 1A–B). Correlation analysis further supported this result, with a strong similarity between Ptbp1 cKO and control astrocytes (ρ = 0.97), and low similarity between Ptbp1 cKO astrocytes and induced neurons (ρ = 0.27) (Figure 6–figure supplement 1C). These findings indicate that, even with PTBP1 loss, cKO astrocytes retain a transcriptional profile very distinct from that of neurons, underscoring that Ptbp1 deficiency alone does not induce astrocyte-to-neuron reprogramming at the transcriptomic level (lines 366-373).

**Reviewer #2 (Public review):**
Summary:The manuscript by Zhang and colleagues describes a study that investigated whether the deletion of PTBP1 in adult astrocytes in mice led to an astrocyte-to-neuron conversion. The study revisited the hypothesis that reduced PTBP1 expression reprogrammed astrocytes to neurons. More than 10 studies have been published on this subject, with contradicting results. Half of the studies supported the hypothesis while the other half did not. The question being addressed is an important one because if the hypothesis is correct, it can lead to exciting therapeutic applications for treating neurodegenerative diseases such as Parkinson's disease.In this study, Zhang and colleagues conducted a conditional mouse knockout study to address the question. They used the Cre-LoxP system to specifically delete PTBP1 in adult astrocytes. Through a series of carefully controlled experiments, including cell lineage tracing, the authors found no evidence for the astrocyte-to-neuron conversion.The authors then carried out a key experiment that none of the previous studies on the subject did: investigating alternative splicing pattern changes in PTBP1-depleted cells using RNA-seq analysis. The idea is to compare the splicing pattern change caused by PTBP1 deletion in astrocytes to what occurs during neurodevelopment. This is an important experiment that will help illuminate whether the astrocyte-to-neuron transition occurred in the system. The result was consistent with that of the cell staining experiments: no significant transition was detected.These experiments demonstrate that, in this experimental setting, PTBT1 deletion in adult astrocytes did not convert the cells to neurons.Strengths:This is a well-designed, elegantly conducted, and clearly described study that addresses an important question. The conclusions provide important information to the field.To this reviewer, this study provided convincing and solid experimental evidence to support the authors' conclusions.Weaknesses:The Discussion in this manuscript is short and can be expanded. Can the authors speculate what led to the contradictory results in the published studies? The current study, in combination with the study published in Cell in 2021 by Wang and colleagues, suggests that observed difference is not caused by the difference of knockdown vs. knockout. Is it possible that other glial cell types are responsible for the transition? If so, what cells? Oligodendrocytes?

We are grateful for the reviewer’s careful reading and valuable suggestions. We have expanded the Discussion to include discussion of possible origins of glial cells responsible for neuronal transition. (lines 441-461)

**Reviewer #1 (Recommendations for the authors):**
(1) Throughout the text and figures, it is customary to write loxP with a capital "P".

We have capitalized “P” in loxP throughout the text and figures.

(2) It would be helpful to indicate the brain regions analyzed above the images in Figure 1B-C, Figure 2A-B, Figure 1 - Supplement 3, and Figure 2 - Supplement 2, as was done in Figure 1 - Supplement 1.

The labels indicating brain regions of corresponding images have been added to the figures.

(3) The arrowheads in Figure 1C, Figure 2B, Figure 3, and several supplemental panels are nearly equilateral triangles, making their direction difficult to discern. Consider using a more slender or indented design (e.g., ➤).

We have replaced triangular arrowheads with indented arrowheads in the figures.

(4) Lines 181-209: This section should be revised, given that the striatum is not a midbrain structure.

We have revised this section to reflect our analysis of the striatum as a brain region of the nigrostriatal pathway rather than a midbrain structure.

**Reviewer #2 (Recommendations for the authors):**
In Supplemental Figure 1, the two open triangles are almost indistinguishable. It would be better if the colors of these open triangles were changed so that it is easier to tell what's what. There is not enough contrast between white and yellow.

We have changed the open triangle arrowheads to solid yellow and violet arrowheads to improve contrast between labels.